# Cross-species standardised cortico-subcortical tractography

Stephania Assimopoulos[1], Shaun Warrington[1], Davide Folloni[2,3],
Katherine Bryant[4,5], Ali-Reza Mohammadi-Nejad[1,6], Wei Tang[7], Saad Jbabdi[8],
Sarah R Heilbronner[9], Rogier B Mars[8,10], Stamatios N Sotiropoulos[1,6]*

[1]Sir Peter Mansfield Imaging Centre, School of Medicine, University of Nottingham,
Nottingham, United Kingdom; [2]Nash Family Department of Neuroscience and
Friedman Brain Institute, Icahn School of Medicine at Mount Sinai, New York, United
States; [3]Lipschultz Center for Cognitive Neuroscience, Icahn School of Medicine
at Mount Sinai, New York, United States; [4]Institute for Language, Cognition, and
the Brain, CNRS, Université Aix-Marseille, Marseille, France; [5]Centre de Recherche
en Psychologie et Neurosciences, UMR 7077, CNRS/Université Aix-Marseille,
Marseille, France; [6]NIHR Nottingham Biomedical Research Centre, Queen's Medical
Centre, University of Nottingham, Nottingham, United Kingdom; [7]Luddy School
of Informatics, Computing and Engineering, Indiana University Bloomington,
Bloomington, United States; [8]Oxford Centre for Integrative Neuroimaging, University
of Oxford, Oxford, United Kingdom; [9]Baylor College of Medicine, Houston, United
States; [10]Donders Institute for Brain, Cognition and Behaviour, Radboud University
Nijmegen, Nijmegen, Netherlands

*For correspondence:
Stamatios.Sotiropoulos@
nottingham.ac.uk

Competing interest: See page
20

Reviewing Editor: Ted D
Satterthwaite, University of
Pennsylvania, United States

## eLife Assessment

This **important** study provides a novel approach for delineating subcortical-cortical white matter bundles. The authors provide **convincing** evidence by harnessing state-of-the-art methods and cross-species data. Together, this effort will be of interest to scientists across multiple subfields and accelerate progress in a biologically critical but methodologically challenging area.

**Abstract** Despite their importance for brain function, cortico-subcortical white matter tracts are under-represented in diffusion magnetic resonance imaging tractography studies. Their non-invasive mapping is more challenging and less explored compared to other major cortico-cortical bundles. We introduce a set of standardised tractography protocols for delineating tracts between the cortex and various deep subcortical structures, including the caudate, putamen, amygdala, thalamus, and hippocampus. To enable comparative studies, our protocols are designed for both human and macaque brains. We demonstrate how tractography reconstructions follow topographical principles obtained from tracers in the macaque and how these translate to humans. We show that the proposed protocols are robust against data quality and preserve aspects of individual variability stemming from family structure in humans. Lastly, we demonstrate the value of these species-matched protocols in mapping homologous grey matter regions in humans and macaques, both in cortex and subcortex.

## Introduction

Function-specific brain activity involves the integration of information from multiple remote brain regions. This integration is enabled by white matter (WM) bundles interconnecting different brain regions (*Passingham et al., 2002*; *Mars et al., 2018a*; *Thiebaut de Schotten and Forkel, 2022*; *Pessoa, 2023*). Of particular interest and importance are bundles connecting cortical areas with deep brain structures. Subcortical structures have important roles in affective, cognitive, motor, and social functions (*Utter and Basso, 2008*; *Bickart et al., 2011*; *Berridge and Kringelbach, 2015*), which emerge through interactions with cortical areas that such connections enable (*Haber, 2016*; *Chumin et al., 2022*; *Bullock et al., 2022*). Hence, the variability of these connections between individuals has been linked to differences in behavioural traits (*Cohen et al., 2009*; *Forstmann et al., 2010*; *Forkel et al., 2022*). Furthermore, their disruption has been associated with abnormal function and pathology in neurodegenerative and mental health disorders (*Haber and Behrens, 2014*; *Heller, 2016*; *Haber et al., 2023*; *Weerasekera et al., 2024*). In the clinic, individual variability in cortico-subcortical connectivity has been used to assist presurgical planning and predict personalised targets for efficacious interventions (*Akram et al., 2017*).

Chemical tracer studies in the non-human primate (NHP) brain have provided, and continue to provide, invaluable insights into cortico-subcortical connections and neuroanatomy in general (*Heilbronner and Chafee, 2019*). Examples include tracing of cortico-striatal connections (*Lehman et al., 2011*; *Heilbronner and Haber, 2014*; *Haber, 2016*; *Safadi et al., 2018*), amygdalofugal (AMF) connections (*Oler et al., 2017*), and thalamo-cortical connections (*Yoshida and Benevento, 1981*; *Lehman et al., 2011*). Comparative neuroanatomy studies can subsequently explore and translate principles of WM organisation learnt from NHPs to humans (*Jbabdi et al., 2013*; *Safadi et al., 2018*; *Folloni et al., 2019*). Brain imaging and, in particular, diffusion magnetic resonance imaging (dMRI) tractography (*Jbabdi et al., 2015*) is a crucial component in these comparative studies and beyond (*Sotiropoulos et al., 2013*; *Assimopoulos et al., 2024*; *Sotiropoulos et al., 2025*), allowing non-invasive mapping of these connections in the living human.

Towards this direction, recent dMRI-based frameworks have been developed to map respective WM bundles (tracts) across NHPs and humans (*Warrington et al., 2020*; *Roumazeilles et al., 2020*; *Bryant et al., 2020*; *Bryant et al., 2021*; *Assimopoulos et al., 2024*). These rely on standardised dMRI tractography protocols, comprising functionally driven, rather than geometric, definitions, enabling automated and generalisable mapping of homologous major WM tracts across species (*Warrington et al., 2020*). These developments have allowed cross-species neuroanatomy studies (*Mars et al., 2018b*; *Mars et al., 2021*) and mapping of connections across humans to study links with brain development, function, and dysfunction (*Thiebaut de Schotten et al., 2020*; *Warrington et al., 2022*). A current limitation of these imaging-based approaches is that they have mainly focused so far on cortico-cortical bundles (with the exception of cortico-thalamic connections).

Tractography protocols for WM bundles that reach deeper subcortical regions, for instance the striatum or the amygdala, are more difficult to standardise. The relative size and proximity of these bundles, and the WM complexities and bottlenecks they go through, can make their mapping through dMRI particularly challenging. As a consequence, considerably fewer studies have proposed solutions for their reproducible reconstruction, both within and across primate species, compared to more major cortico-cortical bundles (*Wassermann et al., 2016*; *Wasserthal et al., 2018*; *Warrington et al., 2020*; *Maffei et al., 2021*). Some existing studies have focused on cortico-striatal bundles (*Forkel et al., 2014*; *Schilling et al., 2020*), uncinate and AMF fasciculi (*Folloni et al., 2019*), parts of the extreme capsule (*Mars et al., 2016*), and the anterior limb of the internal capsule (*Jbabdi et al., 2013*; *Safadi et al., 2018*). However, these either utilise labour-intensive single-subject protocols (*Safadi et al., 2018*; *Folloni et al., 2019*), are not designed to be generalisable across species (*Forkel et al., 2014*; *Schilling et al., 2020*), or are based mostly on geometrically driven parcellations that do not necessarily preserve topographical principles of connections (*Wasserthal et al., 2018*). We propose an approach that addresses these challenges and is automated, standardised, generalisable across two species and includes a larger set of cortico-subcortical bundles than considered before, yielding tractography reconstructions that are driven by neuroanatomical constraints.

Specifically, we build upon our previous work on FSL-XTRACT (*Mars et al., 2018b*; *Warrington et al., 2020*; *Assimopoulos et al., 2024*) to propose standardised protocols and an end-to-end framework for automated cortico-subcortical tractography in the macaque and human brain, considering

**Table 1.** New and revised subcortical protocols.

The developed subcortical tractography protocols for the macaque and human brain. Protocols for anterior commissure, fornix, and uncinate fasciculus were revised from *Warrington et al., 2020*.

| Tract name | Abbreviation |
| --- | --- |
| Amygdalofugal tract | *AMF* |
| Anterior commissure | *AC* |
| Extreme capsule (frontal) | *EmC_f* |
| Extreme capsule (temporal) | *EmC_t* |
| Extreme capsule (parietal) | *EmC_p* |
| Fornix | *FX* |
| Muratoff bundle | *MB* |
| Striatal bundle (sensorimotor) | *StB_m* |
| Striatal bundle (frontal) | *StB_f* |
| Striatal bundle (temporal) | *StB_t* |
| Striatal bundle (parietal) | *StB_p* |
| Uncinate fasciculus | *UF* |

connections between the cortex and the caudate, putamen, and amygdala. To this end, we use prior anatomical knowledge from NHP tracers to define new generalisable protocols, including the AMF tract, the Muratoff bundle (MB) and the striatal bundle (external capsule) with its frontal, sensorimotor, temporal, and parietal parts, augmenting our previous protocols for hippocampal and thalamic tracts (*Warrington et al., 2020*). Due to their close proximity, we also develop new protocols for the respective extreme capsule parts (frontal, temporal, and parietal) and revise previously released protocols (*Warrington et al., 2020*) for the uncinate fasciculus (UF), anterior commissure (AC), and fornix (FX).

We demonstrate the mapping of the respective bundles in the human and macaque brain and show that tractography reconstructions follow topographical principles obtained from tracers. We show that the proposed definitions are robust against dMRI data quality and preserve aspects of individual variability stemming from family structure in humans, as reflected by higher similarity of reconstructed tracts in the brains of monozygotic twins compared to non-twin siblings and unrelated subjects. We subsequently demonstrate how these tractography reconstructions can improve the identification of homologous grey matter (GM) regions across species, both in cortex and subcortex, on the basis of similarity of GM areal connection patterns to the set of proposed WM bundles (*Passingham et al., 2002*; *Mars et al., 2018b*).

## Results

Using prior anatomical knowledge from tracer studies in the macaque, we developed new tractography protocols for the macaque brain and subsequently translated them to the human brain. We considered 23 tracts in total (11 bilateral, 1 commissural), which included tracts connecting the cortex to the amygdala, caudate, and putamen. Specifically, we developed protocols for the *AMF* pathway, the sensorimotor, frontal, temporal, and parietal parts of the striatal bundle/external capsule ($StB_m, StB_f, StB_t, StB_p$), and the *MB*. Due to their proximity, we also developed protocols for the frontal, temporal, and parietal parts of the extreme capsule ($EmC_f$, $EmC_t$, $EmC_p$) (neighbouring to the corresponding external capsule parts), and revised previous protocols for the *UF* (neighbouring to the *AMF*), the *FX* (output tract of the hippocampus next to the amygdala), as well as the *AC* (*Table 1*, *Appendix 1—table 1*).

We used the XTRACT approach (*Warrington et al., 2020*) to define tractography protocols, governed by two principles: (1) protocols are comprised of seed/stop/target/exclusion regions of interest (ROIs) defined in template space, so that they are standardised and generalisable (compared

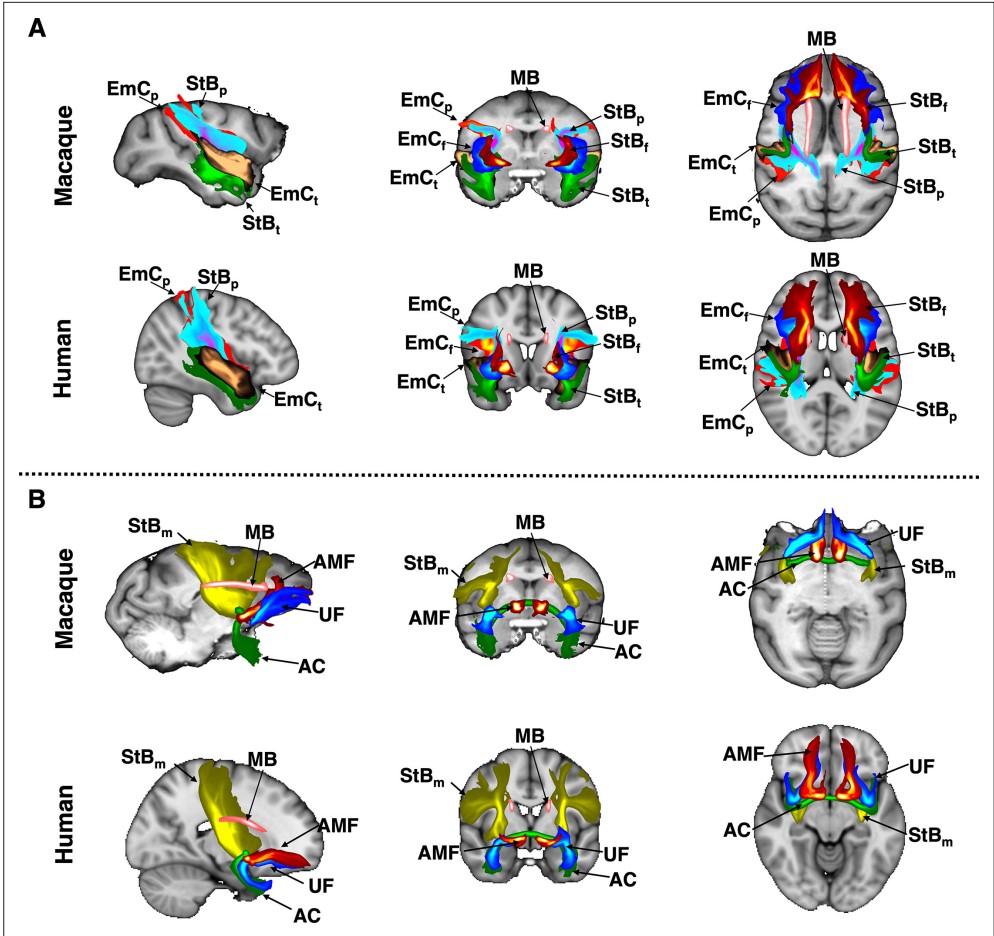

**Figure 1.** Tractography reconstructions of subcortical bundles in the macaque and human brain using correspondingly defined protocols. Maximum intensity projections (MIPs) in sagittal, coronal, and axial views of group-averaged probabilistic path distributions, for all proposed tractography protocols in the macaque (6 animal average) and human (average of 50 subjects from the Human Connectome Project). All MIPs are within a window of 20% of the field of view centred at the displayed slices. (**A**) Frontal, temporal, and parietal parts of the extreme capsule ($EmC_f$, $EmC_t$, $EmC_P$); frontal, temporal, and parietal parts of the striatal bundle ($StB_f$, $StB_t$, $StB_P$); and the Muratoff bundle ($MB$). (**B**) Amygdalofugal tract ($AMF$); anterior commissure ($AC$); uncinate fasciculus ($UF$); sensorimotor part of the striatal bundle ($StB_m$). Path distributions were thresholded at 0.1% before averaging.

to subject-specific protocols), and (2) ROIs are coarse enough and defined equivalently between macaques and humans to enable the tracking of corresponding bundles across species. Full tractography protocols, and modifications to existing protocols, are described in detail in Methods. Protocols were defined in MNI152 template space for human tractography and F99 space (*Van Essen, 2002*; *Glasser and Van Essen, 2011*) for macaque tractography. Additionally, we generalised the macaque protocols to the NIMH Macaque Template (NMT v2) (*Seidlitz et al., 2018*). For clarity, results shown in the main text use the F99 space protocols. Comparison between results in NMT and F99 space can be seen in *Appendix 1—figure 3*.

## Subcortical tract reconstruction across species and comparisons with tracers

Using dMRI data from the macaque ($N = 6$) and human brain ($N = 50$) and the defined protocols, we performed tractography reconstructions for all the tracts of interest. Maximum intensity projections of the resultant group-averaged tract reconstructions for the macaque and human are shown colour-coded in *Figure 1* (individual tracts can be seen in *Appendix 1—figure 1*). These reveal overall

correspondence in the main bodies of tracts across species, while capturing differences in (sub)cortical projections.

We explored whether WM organisation principles known from the tracer literature are captured in these tractography reconstructions (*Bullock et al., 2022*; *Lehman et al., 2011*; *Schmahmann and Pandya, 2006*; *Makris and Pandya, 2009*; *Choi et al., 2017a*; *Liu et al., 2020*). For instance, the striatal bundle (*StB*)/external capsule is always medial to the extreme capsule and the MB runs along the head of the caudate nucleus. *Figure 2A* shows the relative positioning for $StB_f$, $EmC_f$, and $MB$ bundles. Correspondence between tractography results and tract tracing reconstruction in the macaque can be observed, with their relative positions being preserved. This relative position was preserved in the human tractography results as well. Furthermore, the medio-lateral separation is also

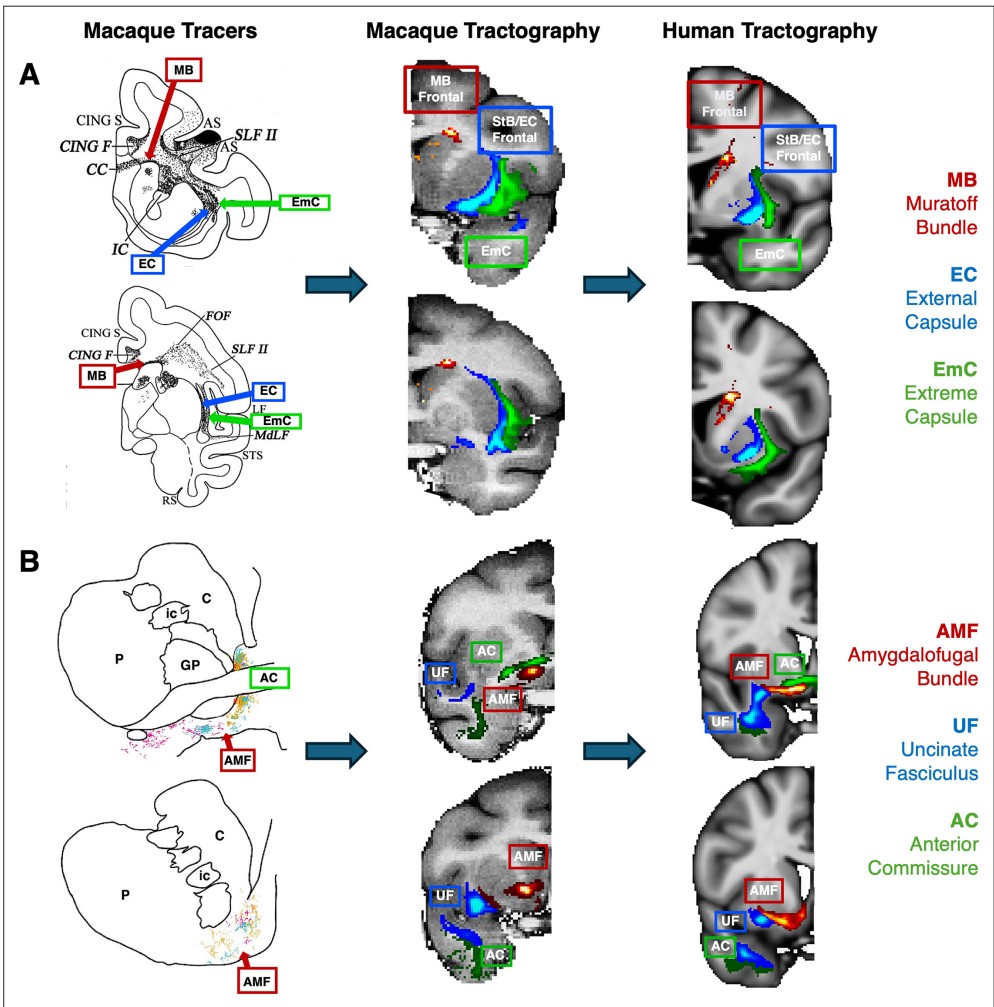

**Figure 2.** Tractography mirrors tracer patterns in the macaque brain, with similar patterns in the human. The proposed protocols were first developed in the macaque guided by tracer literature, and then transferred over to the human. Relative positioning of diffusion magnetic resonance imaging (dMRI)-reconstructed tracts was subsequently explored against the ones suggested by tracers, with good agreement in both species. (**A**) The dorsal–medial/ventral–lateral separation between the extreme an external capsule (here the frontal parts ($EmC_f$) and $StB_f$ shown) is present in macaque tractography, as suggested in the tracer literature. The Muratoff bundle runs along the head of the caudate nucleus. These relative positions are also preserved in the human tractography results. Tracer image modified from *Petrides and Pandya, 2006* with permission (under a CC BY 4.0 licence). (**B**) Similarly for the amygdalofugal (*AMF*) bundle, which runs under the anterior commissure (*AC*) and over the uncinate fasciculus (*UF*), we see agreement with tracer studies with respect to its location in both the macaque and human tractography (*Oler et al., 2017*; *Folloni et al., 2019*; *Oler and Fudge, 2019*). Tracer image adapted from *Oler et al., 2017* with permission (under a CC BY 4.0 licence). In all examples group-average tractography results are shown.

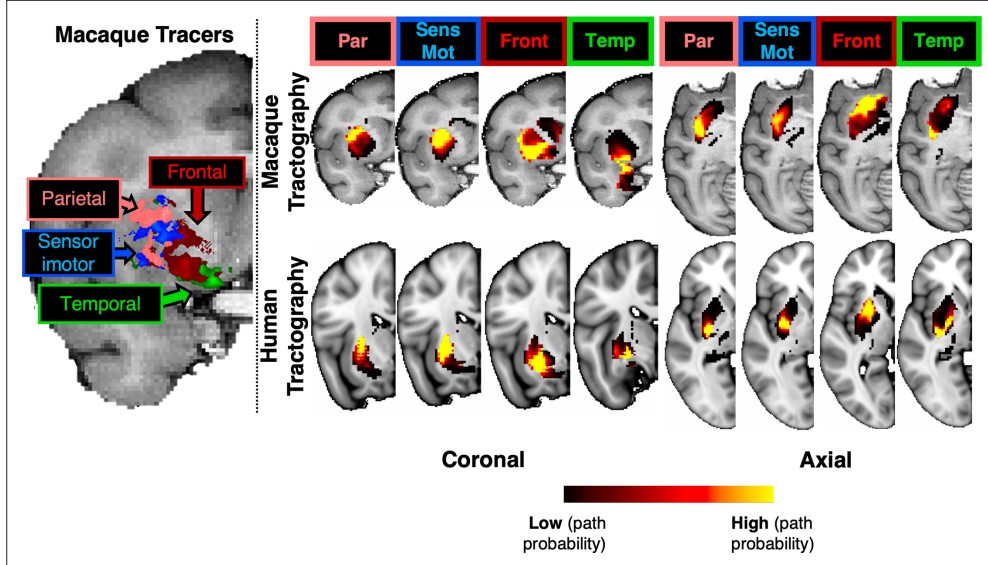

**Figure 3.** Tractography-derived connectivity patterns in the putamen resemble (for both macaque and human) termination sites identified by tracers after injections at different cortical areas (frontal, sensorimotor, parietal, and temporal) in the macaque. Left: Using macaque tracer data from 78 injections in various parts of the cortex, tracer termination sites in the putamen suggested a pattern based on the distinct cortical origin of the tracer injection sites; moving from the dorsolateral to the ventromedial putamen. Right: The path distributions of the different parts of the striatal bundle ($StB_f$, $StBm$, $StB_p$, $StB_t$) within the putamen reveal a similar pattern of connectivity to different parts of the cortex, both for macaque (top) and the human (bottom). Coronal and axial views of group-average results are shown for tractography. Cortical areas (Front: frontal cortex, Par: parietal cortex, Temp: temporal cortex, SensMot: Sensorimotor cortex) were obtained from the CHARM1 parcellation (*Jung et al., 2021*) in the macaque brain (for both tracers and tractography) and from the Harvard parcellation in the human (*Frazier et al., 2005*; *Desikan et al., 2006*; *Makris and Pandya, 2009*).

observed in the other parts of *StB* and *EmC* (i.e. parietal and temporal), as shown in **Appendix 1—figure 2**. Similarly, for the *AMF* bundle (**Figure 2B**), this runs through the AC and ventral pallidum, as well as, in its lateral part, over the *UF*. We see agreement with respect to these relative positions in both the macaque and human.

In addition to the main WM core of the reconstructed bundles, we also explored agreement of the relative connectivity patterns within the striatum between tracers and tractography. Cortical injections of anterograde tracers from different parts of the macaque brain reveal a dorsolateral to ventromedial organisation in the putamen, from parietal to temporal projections (**Figure 3**). Using the path distribution of the tractography reconstructed *StB* parts within the putamen, we could obtain a similar pattern in the macaque brain. This also resembled the pattern found in the human brain, as shown in both coronal and axial views.

## Generalisability across data and individuals

We subsequently explored generalisability and robustness of the tractography protocols against NHP template spaces and dMRI data quality. **Appendix 1—figure 3** shows tract reconstructions in the macaque brain when using the F99 (**Van Essen, 2002**) vs the NMT (**Seidlitz et al., 2018**) templates, similar tractography reconstructions for protocols defined in either of the two templates.

To explore performance against data quality, we compared tractography reconstruction in very high-quality high-resolution data from the Human Connectome Project (HCP) (**Van Essen et al., 2013**; **Sotiropoulos et al., 2013**), to tractography in more standard quality data from the UK Biobank dataset (**Miller et al., 2016**). **Appendix 1—figure 4** demonstrates the ability to reconstruct all tracts across a range of data qualities, with good correspondence of the main bodies of the tracts in both datasets. We quantified this agreement by calculating the mean Pearson's correlation across the set of new and revised tracts for each unique pair of subjects across and within each of the HCP and UK Biobank (UKB) datasets (**Figure 4A**). For reference, we performed similar correlations for the original

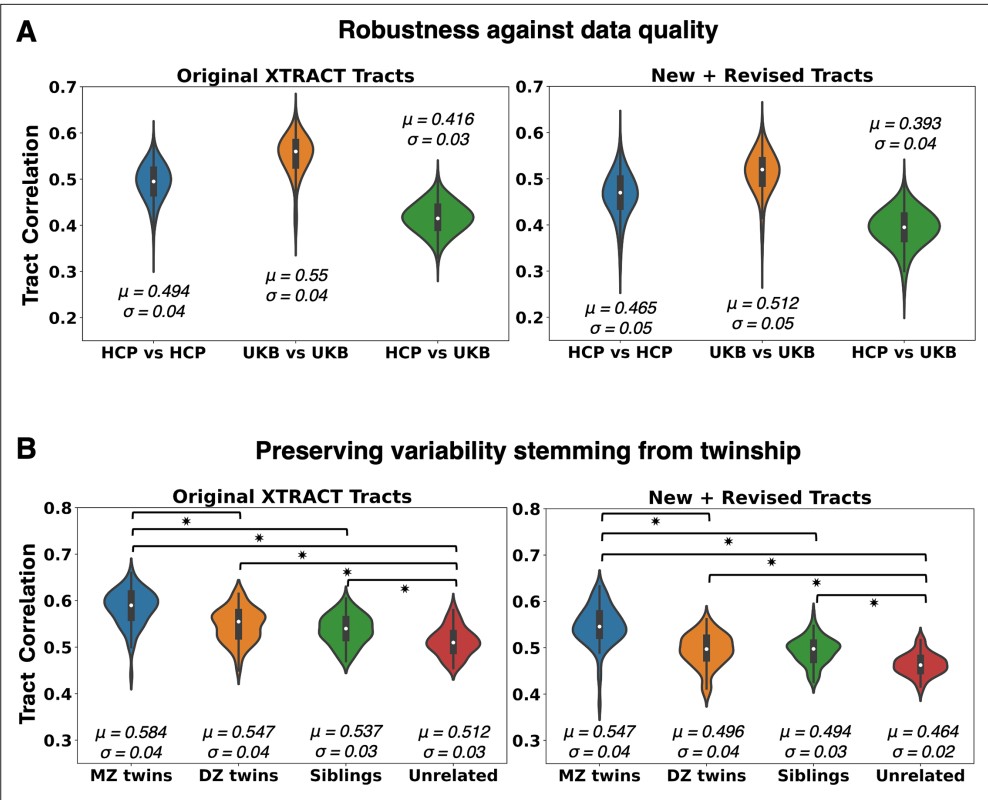

**Figure 4.** Generalisability of proposed tractography reconstructions across data quality and individuals. Results for the new subcortical tracts (right column) are shown against reference corresponding results for the original set of XTRACT tracts (left column), which have been widely used (**Warrington et al., 2020**). (**A**) Tract similarity within and between two in vivo human cohorts, spanning a wide range of diffusion magnetic resonance imaging (dMRI) data quality (HCP: high resolution, long scan time, bespoke setup, UK Biobank (UKB): standard resolution, short scan time, clinical scanner). Violin plots of the average across tracts pairwise Pearson's correlations, between 1225 unique subject pairs within and across the two cohorts, are shown. Correlations are performed on normalised tract density maps with a threshold of 0.5%. Reported $\mu$ is the mean of the correlations across tracts and subject pairs and $\sigma$ is the standard deviation. (**B**) Tract similarity in twins, non-twin siblings, and unrelated subjects. Violin plots of the average across tracts pairwise Pearson's correlations between 72 monozygotic (MZ) twin pairs, 72 dizygotic (DZ) twin pairs, 72 non-twin sibling pairs, and 72 unrelated subject pairs from the Human Connectome Project. Heritable traits are more similar in MZ twins, equally similar in DZ twins and non-twin siblings and more than in unrelated subjects. Asterisk indicates significant pairwise comparisons between groups, as indicated by the brackets.

set of XTRACT tracts (**Warrington et al., 2020**) (see **Appendix 1—table 1** for a list of Original vs New + Revised tracts). Higher correlation was observed within each dataset, but also a sufficiently high correlation between the two datasets. We found similar patterns across datasets both for the original and the new tracts, showcasing that the new protocols behave similarly to the widely used original XTRACT protocols, across data qualities.

The mean agreement between HCP and UKB reconstructions was lower compared to within-dataset agreements. The two cohorts correspond to different age ranges, with HCP having younger adults than the UKB, which could be contributing to these differences. In addition, this was due to occasionally reconstructing a sparser path distribution in the low-resolution data, particularly for some of the new tracts, as both their relative size and their proximity make them more challenging. This highlights the potential importance of having high-resolution data in tracking WM bundles in densely packed areas of higher complexity. It is interesting to note, however, that similar tracts were less/more reproducible between subjects across data qualities. For the lower quality data from the UKB, the tracts with lowest agreement across subjects (**Pessoa, 2023**) were the $AC$ and the temporal part of the extreme capsule ($EmC_t$), while the highest correlations were for the $MB$ and the temporal part of the striatal bundle ($StB_t$). For the higher quality HCP data, the temporal part of the extreme

capsule ($EmC_t$) and the $MB$ were also the tracts with the lowest/highest correlations across subjects, respectively. Hence, certain tract reconstructions were consistently more variable than others across subjects, which may hint at also being more challenging to reconstruct. Taken together, despite differences, our results suggest all tracts could be reconstructed across both data qualities in a generalisable manner (*Appendix 1—figure 4*).

We subsequently explored whether the proposed protocols preserve aspects of individual variability. We used the family structure in the HCP data to explore whether tract reconstructions from monozygotic twin pairs are more similar compared to tracts obtained from other pairs of siblings or unrelated subjects. As shown in *Figure 4B*, we found a decrease in pairwise tract similarity going from monozygotic twins to dizygotic twins and non-twin siblings, and to pairs of unrelated subjects. For reference, we performed the same analysis for the original XTRACT tracts and the same pattern persisted for the new (and revised) tracts, in agreement with previous work (*Bohlken et al., 2014*; *Shen et al., 2014*; *Warrington et al., 2020*). In each analysis, all pairwise differences were significant (Bonferroni corrected $p < 0.05$; following a Mann–Whitney $U$-test), with the exception of dizygotic twins compared to non-twin siblings.

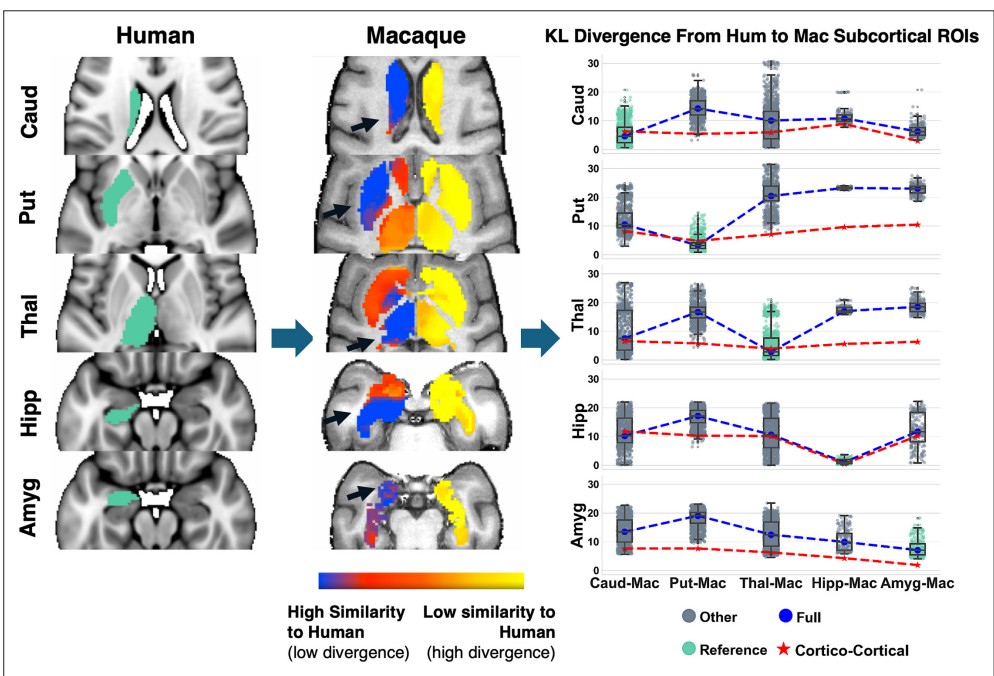

**Figure 5.** Identifying homologous deep brain structures (subcortical nuclei and hippocampus) across species solely by connectivity pattern similarity, obtained by the new tractography reconstructions. Using the corresponding tracts in humans and macaques, connectivity blueprints can be calculated. These are *GMxTracts* matrices, with each row providing the pattern of how a grey matter (GM) location is connected to the predefined set of Tracts (*Mars et al., 2018b*). Left: Starting from the average connectivity blueprints (across the 50 human subjects) of reference human regions of interest (ROIs) (Caud: caudate, Put: putamen, Thal: thalamus, Hipp: hippocampus, Amyg: amygdala), Kullback–Leibler (KL) divergence (or inverse similarity) maps can be computed against the connectivity blueprints of deeper subcortical regions in the macaque (average shown in the middle). The highest connection pattern similarity corresponds to the homologue macaque region of the corresponding human one. Right: Boxplots of KL divergence values between the reference human regions (across the 50 subjects) and the five macaque ones (across the six macaques). Each box shows the quartiles of the data while the whiskers extend to show the rest of the distribution, except for points that are determined to be "outliers". Blue dashed line corresponds to median KL divergence values when all white matter tracts are considered (both cortico-cortical and the new subcortical ones). Red dashed line corresponds to median KL divergence when using only cortico-cortical tracts. When cortico-subcortical tracts are included vs not, there is increased specificity/contrast in the cross-species mapping of these deeper structures. The boxplot with the lowest median divergence is shown in green in each case, indicating the best-matching regions in the macaque to the human reference (i.e. caudate human reference best matches macaque caudate, putamen human reference best matches macaque putamen, etc).

Examples of tract reconstructions on individual subjects are shown in *Appendix 1—figure 6*, *Appendix 1—figure 7*. The figures demonstrate tractography results for subjects corresponding to 10th, 50th, and 90th percentiles of the distribution of tract correlations to the HCP group average. This ranking was also representative of high, medium, and low subject motion across the cohort, respectively. Results demonstrate that the expected patterns are preserved for all tracts ($MB$, $AMF$, $UF$, $EmC$, $StB$ (frontal and parietal parts)). *Appendix 1—figure 6* shows that even the relative medial–lateral organisation of the $StB$ with respect to the $EmC$ is also maintained across the three individual examples, in agreement with the group-average pattern.

## Identifying homologues in cortex and subcortex using tractography patterns

Based on our previous work (*Mars et al., 2018b*), we used the similarity of areal connectivity patterns with respect to equivalently defined WM tracts across the two species to identify homologous GM regions between humans and macaques. With the addition of the new subcortical tracts, we could perform this task for deep brain structures (subcortical nuclei and hippocampus) with considerably greater granularity than before. *Figure 5* demonstrates such identification task for five structures in the left hemisphere (caudate, putamen, thalamus, amygdala, hippocampus) using cortico-cortical and cortico-subcortical tracts (sets of tracts defined in *Appendix 1—table 1*). On the left, the regions in the macaque brain with the lowest divergence (highest similarity) in their connectivity patterns to the connectivity patterns of the corresponding human regions are shown in blue. Using only connectivity pattern similarity, these five structures can be matched almost perfectly across the two species. For instance, human putamen (left hemisphere) has more similar connectivity (lower divergence) to macaque putamen (left hemisphere), human thalamus (left hemisphere) to macaque thalamus (left hemisphere), etc. Since we are mapping structures in the left hemisphere using left hemisphere tracts, we observe a low similarity in the contralateral (right) hemisphere, as expected. On the right of *Figure 5*, this identification is quantified even further, highlighting the value of considering the new tracts. For every human left-hemisphere region (specified on the vertical axis), the boxplot of divergence of connectivity patterns to each of the five macaque deep brain regions (left hemisphere) is plotted. The best match corresponds to the boxplot with the lowest values (green) and the dashed blue lines show the medians of these boxplots for each case. For reference, the medians of the divergence values when not considering the new subcortical tracts are shown with the red dashed lines, which are overall more flat (with the exception of the hippocampus which has a connectivity pattern strongly driven by the dorsal subsection of the cingulum bundle ($CBD$), a cortico-cortical tract). It is evident that considering the new tracts provides enhanced contrast between the subcortical structures' connectivity patterns, enabling their correct identification. The improvement is thus not in the best match, but in the specificity of the match.

Having shown increased contrast and specificity in the mapping of deep brain structures, we investigated whether we see a similar effect in the cortex. We selected a set of nearby frontal region pairs to map across the human and the macaque (*Figure 6*), since a number of the new tracts connect frontal regions to the subcortex. Specifically, we considered the dorsomedial prefrontal cortex ($dmPFC$), the ventromedial prefrontal cortex ($vmPFC$), the rostral orbitofrontal cortex ($OFC_r$), and the frontal operculum ($FOp$). These regions were also chosen as they are part of different functional networks (default mode, limbic, and frontoparietal networks), equivalently defined between the macaque and human (*Thomas Yeo et al., 2011*).

The prediction from human to macaque is shown in *Figure 6A*, while the converse prediction from macaque to human is shown in *Figure 6B*. In each case, we compare the prediction using only cortico-cortical tracts (column 1), using only cortico-subcortical tracts (column 2), and using the full set (column 3) (sets of tracts defined in *Appendix 1—table 1* – the middle cerebellar peduncle ($MCP$) was not used in these comparisons). The predicted areas with the highest similarity in connectivity patterns are depicted in blue, while the a priori expected homologue region borders have been outlined in cyan. These results demonstrate benefits when using the subcortical tracts, with mapping of some regions (for instance $vmPFC$ and $OFC_r$) being improved more than others. However, in general, we observed an increase in cross-species similarity in the corresponding areas of interest, combined with a decrease in similarity everywhere else in the cortex, when we considered cortico-subcortical tracts (columns 2 and 3) compared to when we considered cortico-cortical tracts alone (column 1).

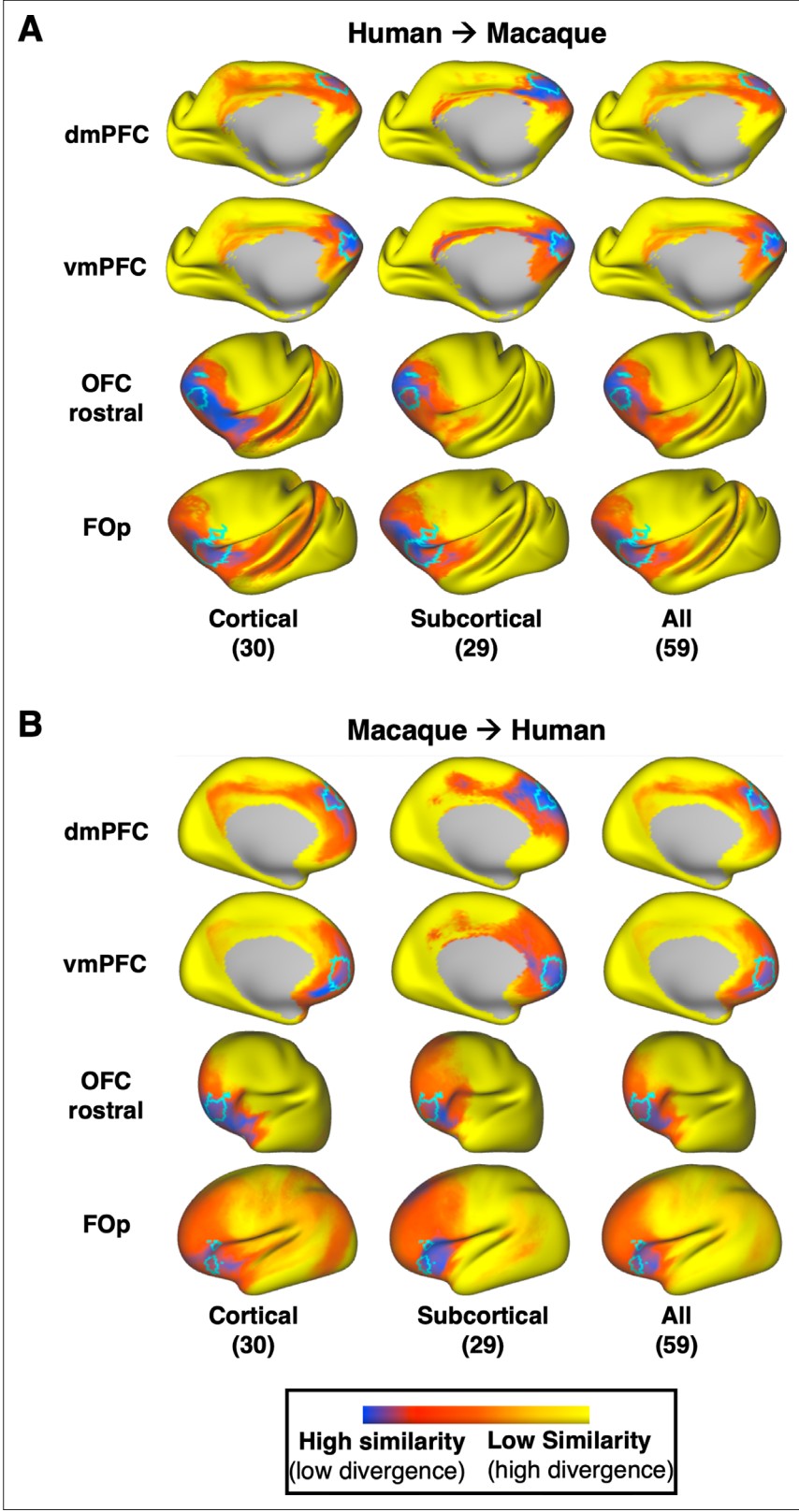

**Figure 6.** Identifying homologous cortical regions across species solely by connectivity pattern similarity, obtained with and without the new tractography reconstructions. Two pairs of neighbouring frontal regions were chosen (*dmPFC*: dorsomedial prefrontal cortex and *vmPFC*: ventromedial prefrontal cortex, *OFC_R*: rostral orbitofrontal cortex and *FO_P*: frontal operculum) and their mapping from human to macaque (**A**) and from macaque to human

*Figure 6 continued on next page*

*Figure 6 continued*

(**B**) was explored. For comparison, we overlay in cyan the corresponding homologue regions in each species, as defined in *Folloni et al., 2019*. (**A**) Kullback–Leibler (KL) divergence maps in the macaque for a given human cortical reference region (one region per row), representing the similarity in connectivity patterns across the macaque cortex to the average pattern of the human reference region. KL divergence maps are calculated using cortico-cortical (first column), cortico-subcortical (second column), and all tracts (third column) to highlight the effect of the cortico-subcortical tractography reconstructions in the prediction. Subcortical tracts provide larger benefits for the prediction of *vmPFC* and *OFC$_r$*, increasing specificity with respect to the expected borders. (**B**) Same as in A, but using macaque regions as reference and making predictions on the human cortex. KL divergence maps in the human for a given macaque cortical region, representing the similarity of connectivity pattern across the human cortex to the average pattern of the reference macaque region. Overall, in both species, an increased similarity to the reference regions in the homologue areas and decreased similarity across the rest of the cortex is observed, when cortico-subcortical tracts are considered (second or third column). Using the average human (across 50 subjects) and average macaque (across 6 animals) blueprints for this analysis.

*Figure 7* provides a further insight into these mappings, by plotting the connectivity patterns of the human regions against the pattern of its identified best match in the macaque brain. As can be observed, despite the relative proximity of these frontal regions, we have distinct patterns across them. With the exception of *dmPFC*, the connectivity patterns of all other regions have major contributions from the frontal striatal bundle, the extreme capsule and the *AMF* tract and connection patterns

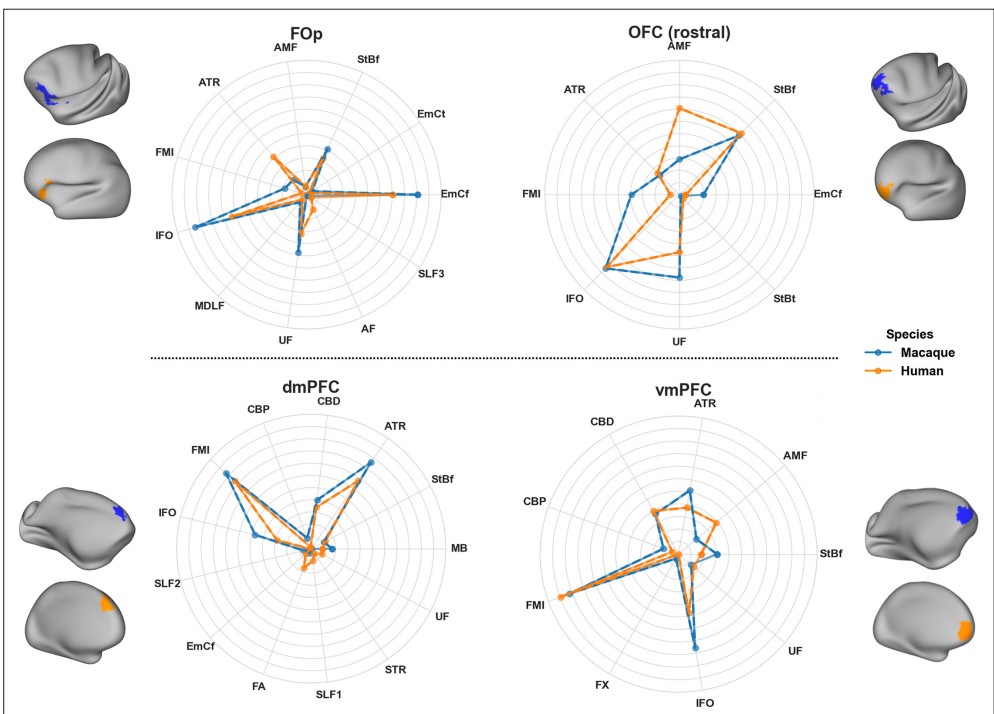

**Figure 7.** Connectivity patterns for neighbouring frontal region pairs, showing distinct cortico-subcortical tract contributions in macaque and human. Considered regions are the same as in *Figure 6A*, that is, *FO$_P$*: frontal operculum, *OFC$_r$*: rostral orbitofrontal cortex, *dmPFC*: dorsomedial prefrontal cortex, *vmPFC*: ventromedial prefrontal cortex. Reference regions were chosen in the human cortex, shown in orange, and obtained from *Folloni et al., 2019*. The best matching region across the whole macaque cortex was identified by the minimum Kullback–Leibler (KL) divergence in connectivity patterns (thresholded at the 7th percentile in each case) and is shown in blue. Average connectivity patterns for the reference and best-matching regions are depicted using the polar plots. For each region, similarities in the connectivity patterns between the macaque and human can be observed, with the new cortico-subcortical bundles contributing to these patterns. For instance, *FO$_P$* has a strong connection pattern involving *EMC$_f$* and uncinate fasciculus (UF) and moderately *StB$_f$* and AF, while its neighbouring *OFC$_r$* has a stronger pattern involving *StB$_f$* and *UF*, compared to *EMC$_f$*. These differences are preserved across both species. Using the average human (across 50 subjects) and average macaque (across 6 animals) blueprints for this analysis.

to these cortico-subcortical bundles enable better separation of these nearby regions. For instance, *vmPFC* and *dmPFC* have both connections through the cingulum bundle, the corpus callosum and the inferior fronto-occipital fasciculus. However, they connect differently to the striatal bundle, the AMF tract and the anterior thalamic radiation, and the addition of these tracts in the connectivity patterns allows the two regions to be better distinguished. The *FOp* and *OFC*$_r$ have both relatively strong connection patterns to the uncinate and the inferior fronto-occipital fasciculi, but it is their different pattern of connections to extreme and external capsules and the AMF tract that enable their better separation.

## Discussion

We introduced standardised dMRI tractography protocols for delineating cortico-subcortical connections between cortex and the amygdala, caudate, putamen, and the hippocampus, across humans and macaques. Building upon our previous work (*Mars et al., 2018b*; *Warrington et al., 2020*), which already provided protocols for cortico-thalamic radiations, and guided by the chemical tracer literature in the macaque, we devised the new protocols first for the macaque and then extended to humans. We demonstrated that our reconstructed tracts preserve topographical organisation principles, as suggested by tracers (*Haber et al., 2006*; *Haber, 2016*; *Oldham and Ball, 2023*).

As outlined in *Schilling et al., 2020*, tractography reconstructions can be highly accurate if information about where pathways go, and where they do not go is available. This is the philosophy behind the proposed protocols, which provide this type of constraints across different bundles. At the same time, these constraints are relatively coarse so that they are species-generalisable. We found that the proposed approaches yield tractography reconstruction across a range of datasets and respect individual similarities stemming from twinship. We further assessed the efficacy of these protocols in performing connectivity-based identification of homologous cortical and subcortical areas across the two species (*Mars et al., 2018b*; *Mars et al., 2021*; *Warrington et al., 2022*).

Mapping WM tracts that link cortical areas with deep brain structures (subcortical nuclei and hippocampus), as done here, enhances capabilities for studying neuroanatomy in many contexts, from evolution and development to mental health and neuropathology. As one of the (evolutionarily) older brain structures, the subcortex modulates brain functions including basic emotions, motivation, and movement control, providing a foundation upon which the more complex cognitive abilities of the cortex could develop and evolve (*Haber et al., 2006*; *Pennartz et al., 2009*; *Haber, 2016*; *Sherman, 2016*; *Cruz et al., 2023*). This modulatory function is mediated via WM bundles (*Haber, 2016*; *Chumin et al., 2022*). Consequently, their disruption is linked to abnormal function and pathology, in mental health, neurodegenerative, and neurodevelopmental disorders (*Heller, 2016*; *Peters et al., 2016*; *Weerasekera et al., 2024*). For example, in depression, fronto-thalamic (*Bhatia et al., 2018*), cortico-amygdalar (*Arnsten and Rubia, 2012*; *Jalbrzikowski et al., 2017*), and cortico-striatal (*van Velzen et al., 2020*) connectivity changes have been reported, while in schizophrenia there are associated fronto-striatal (*Levitt et al., 2017*) and hippocampal connectivity (*Ikeda et al., 2023*) changes. In Parkinson's, there is impairment in fronto-striatal connectivity (*Theilmann et al., 2013*; *Von Der Heide et al., 2013*; *Khan et al., 2019*; *Marecek et al., 2024*), while fronto-thalamic and cingulate connectivity are impaired in Alzheimer's disease (*Von Der Heide et al., 2013*; *Bubb et al., 2018*). Connectivity between the frontal lobe and the amygdala, thalamus, and striatum, as well as cingulum connectivity, is impaired in obsessive compulsive disorder, autism spectrum disorder, and attention deficit hyperactivity disorder (*Langen et al., 2012*; *Arnsten and Rubia, 2012*; *Haber and Behrens, 2014*; *Kilroy et al., 2022*). Therefore, reconstructing connectivity of these deep brain structures (striatum, thalamus, amygdala, and hippocampus) in a standardised manner, as enabled by our proposed tools, allows for further investigation into a wide range of disorders.

In addition, tractography of connections linking to/from deep brain structures has been used or proposed for guiding neuromodulation interventions, for example, deep brain stimulation (DBS) (*Haber et al., 2021*; *Alagapan et al., 2023*) or repetitive transcranial magnetic stimulation (rTMS) (*Peters et al., 2016*). DBS can inherently target subcortical structures and connectivity of subcortical circuits can be used to identify efficacious stimulation targets (*Pouratian et al., 2011*; *Akram et al., 2017*). rTMS, on the other hand, modulates subcortical function indirectly by targeting the structurally connected cortical areas. For example, dmPFC has been targeted to modulate the reward circuitry, in cases of anhedonia, negative symptoms in schizophrenia and major depression disorder (*Dunlop*

et al., 2020; Gan et al., 2021; Bodén et al., 2021), while the vmPFC has been used as a target to modulate the prefrontal–striatal network (part of the limbic system) and regulate emotional arousal/anxiety (Chen et al., 2020; Kroker et al., 2022; Moses et al., 2025). Our results show a good mapping across species of both these cortical regions with specificity in their connectional patterns. Additionally, the motor cortex has been used as a target to modulate cortico-striatal connectivity in general anxiety disorder (Balderston et al., 2020; Fitzsimmons et al., 2024). We thus anticipate that having a standardised set of tracts linking the striatum, the hippocampus, the amygdala and the thalamus (all potential sites for stimulation) to specific cortical areas can assist the planning of interventions.

Our cross-species approach naturally lends itself to the study of evolutionary diversity. A number of comparative studies have revealed differences and similarities when comparing brain connectivity between humans and non-human primates (Barrett et al., 2020), including macaques (Mars et al., 2018b; Warrington et al., 2022) and chimpanzees (Bryant et al., 2025). Our work naturally extends these efforts and provides new tools for studying this diversity in deeper structures and subcortical nuclei. The ever-increasing availability of comparative MRI data (Bryant et al., 2021; Tendler et al., 2022) allows the definition of similar protocols in more species, such as the gibbon (Bryant et al., 2020; Bryant et al., 2024) or the marmoset monkey, and even across geometrically diverse brains depicting different stages of neurodevelopment (e.g. neonates vs adults) enabling concurrent studies of phylogeny and ontogeny (Warrington et al., 2022).

Our protocols have been developed and tested using FSL-XTRACT, but, in principle, are not specific to FSL. We have not evaluated performance with other tools, but these standard-space protocols could be translated into other tractography approaches. As described before, the protocols are recipes with anatomical constraints, including regions to which the corresponding WM pathways connect and regions they do not, constructed with cross-species generalisability in mind. Caution may be needed, however, if applying such protocols for segmenting whole-brain tractograms, as these can induce more false positives than tractography reconstructions from smaller seed regions and may require stricter exclusions.

Despite the potential demonstrated in this work, our study has limitations. As this is the first endeavour of this scale to map cortico-subcortical connections in a standardised manner and across two species, it is not exhaustive. Tracts linking the cortex to the striatum were prioritised as they are of increased relevance in human development and disease. However, expanding to include more tracts targeting other structures would provide a more holistic view. Our protocols were developed in the adult human brain. Future work will translate them to the infant brain (expanding on previous work Warrington et al., 2022) to interrogate cortico-subcortical connectivity across development. Tractography validation is a challenge, as is validation for any indirect and non-invasive imaging approach. We explored and demonstrated the generalisability of the proposed protocols, both within and across species. We also showed how the imaging-based reconstructions follow topographical organisation principles suggested by tracers.

# Materials and methods

**Key resources table** Resources used in this work.

| Reagent type (species) or resource | Designation | Source or reference | Identifiers | Additional information |
|---|---|---|---|---|
| Software, algorithm | BEDPOSTX | Jbabdi et al., 2012; Hernández et al., 2013 | BEDPOSTX | FSL package |
| Software, algorithm | XTRACT | Warrington et al., 2020 | XTRACT | FSL package |
| Software, algorithm | PROBTRACKX | Behrens et al., 2007; Hernandez-Fernandez et al., 2019 | PROBTRACKX | FSL package |
| Software, algorithm | FNIRT | Andersson et al., 2007 | FNIRT | FSL package |
| Software, algorithm | RheMap | Jouandet and Gazzaniga, 1979 | RheMap | |

*Continued on next page*

*Continued*

| Reagent type (species) or resource | Designation | Source or reference | Identifiers | Additional information |
|---|---|---|---|---|
| Software, algorithm | Python 3.12.11 | | Python | General Analysis (FSL package) |
| Software, algorithm | Connectome Workbench v2.1.0 | URL | Workbench | General Analysis |

## Tractography protocols

Guided by tract tracing and neuroanatomy literature, we devised tractography protocols for 18 subcortical bundles (nine bilateral – *Table 1*) using the XTRACT approach (*Warrington et al., 2020*). We also revised protocols for three more bundles (two bilateral, one commissural), compared to their original version (*Warrington et al., 2020*). All protocols followed two principles: (1) comprised of seed/stop/target/exclusion ROIs defined in template space, so that they are standardised and generalisable, and (2) ROIs defined equivalently between macaques and humans to enable tracking of corresponding bundles across species. The human protocols were defined in MNI152 space. The macaque protocols were defined in F99 and also in NMT space.

The tracts included the *AMF* tract, the *UF*, *AC*, sensorimotor, temporal, parietal, and frontal parts of the striatal bundle (*StB*)/external capsule (*EC*), *MB*/subcallosal fasciculus, as well as the extreme capsule (*EmC*) parts that run close to the putamen connecting the insula to the frontal, temporal, and parietal cortices. All XTRACT tracts (Original, Revised, and New) are summarised in *Appendix 1—table 1* .

Detailed protocol definitions are presented below and summarised in *Figure 8* (for completeness, the previously published thalamic radiations from *Warrington et al., 2020* are presented in Appendix 1 Materials). Briefly, the *AMF* and *UF* protocols are a standard-space generalisation of the individual subject-level protocols presented in *Folloni et al., 2019*. For the remaining protocols, we first devised them in the macaque guided by tract tracer literature. Specifically, the approach we took was to first identify anatomical constraints from neuroanatomy literature for each tract of interest independently, derive and test these protocols in the macaque. Thus, each devised protocol included a unique combination of anatomically defined masks (based on literature descriptions of the tracts), delineated in standard macaque space (F99). We then developed corresponding protocols in the human using correspondingly defined landmarks (delineated in standard MNI space). We optimised in an iterative fashion based on two criteria: (1) the protocols generalise well to humans, and (2) when considering groups of bundles, the generated reconstructions follow topographical principles known from tract tracing literature.

We modified existing XTRACT protocols to improve their specificity in the subcortex. Specifically, we developed a new *UF* protocol based on the protocol presented in *Folloni et al., 2019*. We also modified the *AC* protocol to improve temporal lobe projections and slightly enhance projections to the amygdala, and modified the *FX* one to reduce amygdala projections by placing an exclusion in the amygdala.

Given the proximity of the newly defined tracts, we evaluated the new protocols against their ability to capture patterns known from the tracer literature. These included relative positioning of each tract with respect to neighbouring tracts (*Figure 2*) and topographical organisation of certain bundle terminals within subcortical nuclei (*Figure 3*).

### New protocol definitions

### AMF pathway

We derived generalisable template-space protocols to reconstruct the limbic-cortical ventral AMF pathway, following the subject-specific protocols in *Folloni et al., 2019*. The AMF pathway courses between the amygdala and the prefrontal cortex (PFC), running alongside the *UF* medially and finally merging with the *UF* in the posterior orbitofrontal cortex (OFC). As in *Folloni et al., 2019*, the seed included voxels with high fractional anisotropy in an anterior–posterior direction in the sub-commissural WM. We used a target covering all brain at the level of caudal genu of the corpus callosum (same target as in the revised *UF* protocol, described further below). Exclusions include an axial plane through the *UF*, the internal and external capsules, the corpus callosum, the cingulate, the

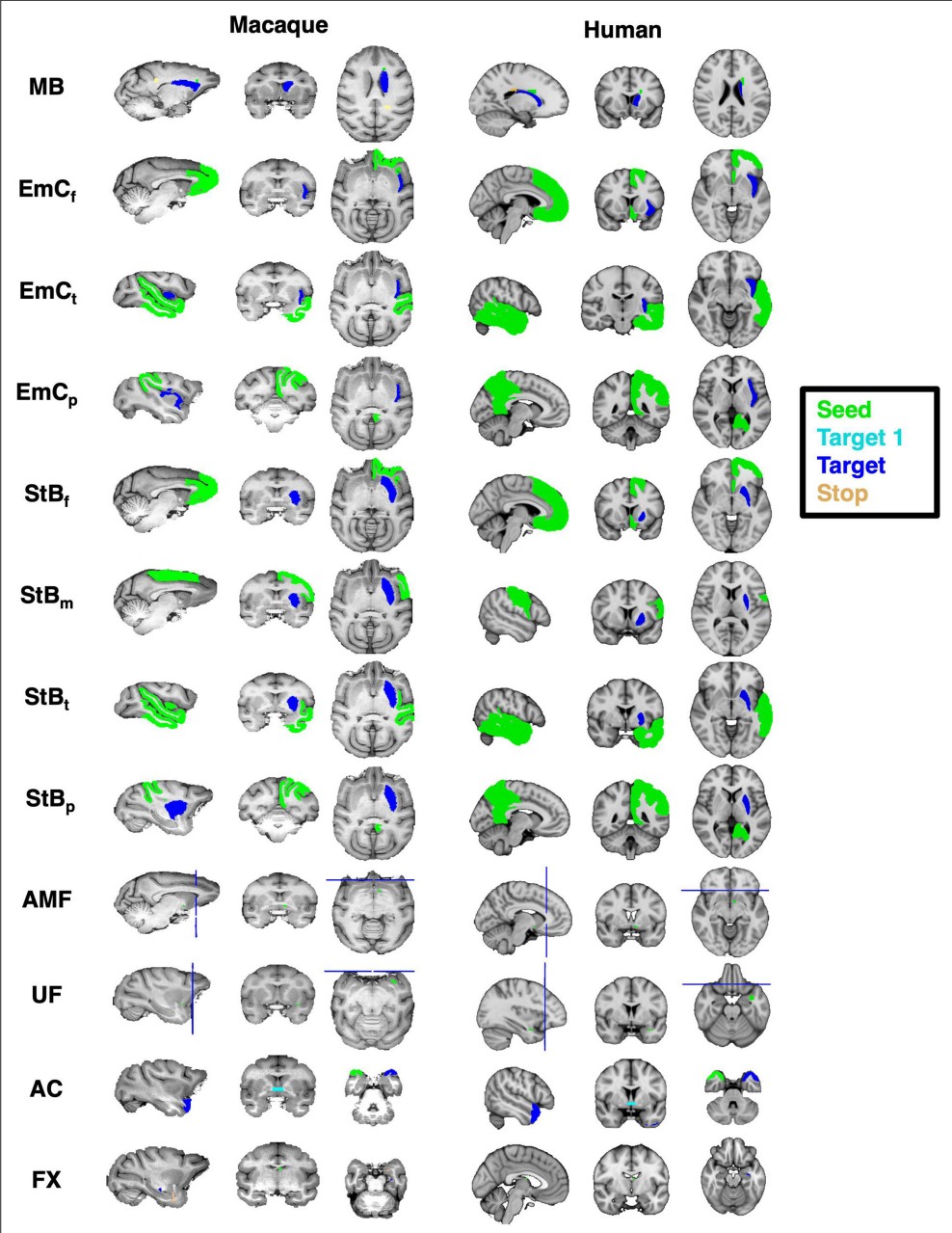

**Figure 8.** Corresponding tract protocol definitions across species. Protocol definitions for all new (and revised) tracts in the human and macaque. Protocols were first designed in the macaque brain guided by macaque tracer literature, and then transferred over to the human. Colour-coded regions depict the seed, target, and stop masks. Exclusion masks are not shown for ease of visualisation.

Sylvian fissure, the AC, the FX, and a large coronal exclusion covering all brain dorsal to the corpus callosum and extending inferiorly to the frontal operculum and insula at the level of the middle frontal gyrus.

## Striatal bundle ($StB_f$, $StB_m$, $StB_p$, and $StB_t$)

The striatal bundle ($StB$) is a bundle system that connects the cortex to the striatum and joins the external capsule ($EC$). Although terminations reach both the caudate and the putamen, they primarily terminate in the putamen (*Schmahmann and Pandya, 2006*; *Makris and Pandya, 2009*; *Bullock et al., 2022*). Here, we defined protocols for bundles that connect the putamen with frontal (including

anterior cingulate) lobe, sensorimotor cortex, parietal, and temporal lobes. For all parts, we used the putamen as the target. For $StB_f$, the OFC, PFC, ACC, and frontal pole made up the seed. For $StB_m$, the primary sensorimotor cortex (M1–S1) was used as seeds. For $StB_t$ and $StB_p$, the temporal and parietal lobes were, respectively, used as a seed. The exclusion masks shared many commonalities but also had differences. For all parts, exclusions included a midsagittal plane, the subcortex, except for the putamen, as well as the occipital lobe. For each of the parts, we additionally excluded the seeds for every other $StB$ bundle.

### Muratoff bundle

The subcallosal fasciculus tract (as called in human neuroanatomy) or MB (as called in non-human animal neuroanatomy) (*Schmahmann and Pandya, 2006*; *Liu et al., 2020*) is a complex system of projection fibres which runs beneath the corpus callosum, above the caudate nucleus at the corner formed by the internal capsule and the corpus callosum (*Forkel et al., 2014*). Although terminations reach both the caudate and the putamen, they primarily terminate in the caudate head (*Schmahmann and Pandya, 2006*; *Forkel et al., 2014*; *Liu et al., 2020*). As its cortical projections are challenging to capture and isolate using tractography, we defined a protocol for the major core of the bundle. We used a seed in the WM adjacent to the caudate head and a target in the WM adjacent to the caudate tail. A stop mask was used beyond the target in the WM above the target. Exclusions included the contralateral hemisphere, the subcortex (except for the caudate head), the brainstem, the parietal, occipital, frontal, and temporal cortices.

### Extreme capsule ($EmC_f$, $EmC_p$, $EmC_t$)

The extreme capsule is a major association fascicle that carries association fibres between frontal–temporal and frontal–parietal, as well as these areas and the insula (*Makris and Pandya, 2009*). It lies between the claustrum and the insula, with the claustrum being considered the boundary between the $EmC$ and the $EC$ (*Bullock et al., 2022*). We defined protocols connecting the insula to frontal, parietal, and temporal cortices. For all parts, we used the insula as the target, while for seeds, we used the same seeds as for the corresponding $StB$ parts. Hence, for the $EmC_f$ protocol, the frontal pole was used as the seed. For $EmC_p$, the parietal lobe was a seed, and for $EmC_t$ the temporal lobe was a seed. Exclusions for all $EmC$ parts included the contralateral part of the brain, the subcortex, as well as the occipital lobe, and lateral parts of the somatosensory and motor cortices. In addition, for each subdivision, the exclusion mask also included the seed mask for every other subdivision.

## Revisions to previous XTRACT protocols

### Uncinate fasciculus

The $UF$ lies at the bottom part of the extreme capsule, curving from the inferior frontal cortex to the anterior temporal cortex. Given the neighbouring bundles that were newly defined, we took a new approach to the $UF$ compared to the original XTRACT implementation (*Warrington et al., 2020*), now following the principles of *Folloni et al., 2019*. Briefly, we used an axial seed in the WM rostro-laterally to the amygdala in the anterior temporal lobe. A target covered all brain at the level of the caudal genu of the corpus callosum. Exclusions included the basal ganglia, a coronal plane posterior to the seed, the corpus callosum, the cingulate, the Sylvian fissure, the AC, and a large coronal exclusion covering all brain dorsal to the corpus callosum and extending inferiorly to the frontal operculum and insula at the level of the middle frontal gyrus. This implementation provided improved connectivity to the dorsal frontal cortex and aided separability with respect to neighbouring WM bundles.

### Anterior commissure

Compared to original XTRACT protocol, we entirely re-worked the $AC$ protocol. Previously, the midline main body of the $AC$ was the seed with targets either side and stops at the amygdala. Now, we use a temporal pole as the seed, the main body of the $AC$ as a waypoint and the contralateral temporal pole as the final target. For the human, we use the Harvard-Oxford temporal pole ROI (*Desikan et al., 2006*). For the macaque, we use the CHARM temporal pole ROI (*Jung et al., 2021*). The temporal pole seed/target pair is flipped and tractography is repeated, taking the average of runs. Compared to the previous version, this protocol provides greater symmetry in resultant reconstructions and greater connectivity to the poles of the temporal cortex, as suggested in the literature

(*Jouandet and Gazzaniga, 1979*; *Catani and Thiebaut de Schotten, 2013*; *Fenlon et al., 2021*; *Bullock et al., 2022*), and slightly enhanced connectivity to the amygdala.

## Fornix

For the $FX$, a main output tract of the hippocampus, we have added an exclusion mask to the amygdala to prevent $FX$ leakage to the amygdala, thus providing a cleaner $FX$ compared to the original XTRACT implementation (*Warrington et al., 2020*). For the human protocol, we used the Harvard-Oxford amygdala ROI (*Frazier et al., 2005*). For the macaque, we used the SARM amygdala ROI (*Hartig et al., 2021*).

## Data

### Macaque MRI data

We used six high-quality ex vivo rhesus macaque dMRI datasets, available from PRIME-DE (*Milham et al., 2018*). As described in *Mars et al., 2018b*; *Warrington et al., 2020*, these were acquired using a 7T Agilent DirectDrive console, with a 2D diffusion-weighted spin-echo protocol with single-line readout protocol with 16 volumes acquired at $b = 0$ s/mm$^2$, 128 volumes acquired at $b = 4000$ s/mm$^2$, and a 0.6-mm isotropic spatial resolution.

### Human MRI data

We used high quality minimally preprocessed (*Glasser et al., 2013*) in vivo dMRI data from the young adult HCP (*Van Essen et al., 2013*; *Sotiropoulos et al., 2013*). The HCP data were acquired using a bespoke 3T Connectom Skyra (Siemens, Erlangen) with a monopolar diffusion-weighted (Stejskal–Tanner) spin-echo echo planar imaging sequence with an isotropic spatial resolution of 1.25 mm, three shells ($b$ values = 1000, 2000, and 3000 s/mm$^2$), and 90 unique diffusion directions per shell plus 6 $b = 0$ s/mm$^2$ volumes, acquired twice with opposing phase encoding polarities. Data correspond to total scan time per subject of approximately 55 min. For this study, we randomly drew 50 HCP subjects (age range, 22–36 years of age, 24/26 females/males).

To assess robustness against data quality, we also used data from the UK Biobank (3T Prisma, 32 channel coil, 2 mm isotropic resolution, $b$ values = 1000 and 2000 s/mm$^2$, 50 directions per shell). The UK Biobank data are acquired with approximately 6.5 min scan time per subject and therefore represent more standard quality datasets, achievable in a clinical scanner (*Miller et al., 2016*). Fifty subjects were randomly drawn from the UK Biobank (UKB) (age range 42–65 years of age, 31/19 females/males). For both HCP and UKB cohorts, we ensured that the distribution of QC metrics (such as subject motion and image SNR/CNR) was representative of the full HCP and UKB cohorts that we had available.

### Macaque tracer data

Tracer data were used to test aspects of the striatal bundle protocols (*Figure 3*). These were made available by SRH and were obtained from an existing collection of injections in 19 macaque brains, from *Weber and Yin, 1984*; *Baizer et al., 1993*; *Yeterian and Pandya, 1995*; *Yeterian and Pandya, 1998*; *Ferry et al., 2000*; *Haber et al., 2006*; *Parvizi et al., 2006*; *Schmahmann and Pandya, 2006*; *Calzavara et al., 2007*; *Choi et al., 2017a*; *Choi et al., 2017b* and cases from the laboratory of SRH. Specifically, anterograde tracers were injected across 78 cortical locations, and their terminations within the putamen were recorded in coronal slices of the NMT template space at 0.5 mm resolution. Specifically, the injection sites were first assigned to one of four cortical ROIs (frontal, parietal, temporal, and sensorimotor cortices), obtained from the NMT CHARM v1 parcellation (*Jung et al., 2021*). For each of these four injection ROIs, we counted all the corresponding terminations within the putamen, and then divided by the total number of termination sites. This resulted in a termination probability map for each cortical region across the putamen, and these termination maps were smoothed using spline interpolation. The putamen mask was obtained from the NMT SARM v1 parcellation (*Hartig et al., 2021*). To compare against tractography in F99 space, these maps were nonlinearly registered from NMT to F99 space using RheMAP (*Klink and Sirmpilatze, 2020*).

## MRI data preprocessing

### Crossing fibre modelling and tractography

For both the human and macaque data, we modelled fibre orientations for up to three orientations per voxel using FSL's BEDPOSTX (*Jbabdi et al., 2012*; *Hernández et al., 2013*) (Key resources table). These orientations were used in tractography. Probabilistic tractography was performed using FSL's XTRACT (*Warrington et al., 2020*), which uses FSL's PROBTRACKX (*Behrens et al., 2007*; *Hernandez-Fernandez et al., 2019*) (Key resources table). The standard-space protocol masks were used to seed and guide tractography, which occurred in diffusion space for each dataset. 60 major WM fibre bundles were reconstructed 30 cortico-cortical, 29 cortico-subcortical, 1 cerebellar, *Appendix 1—table 1* . A curvature threshold of 80° was used, the maximum number of streamline steps was 2000, and subsidiary fibres were considered above a volume fraction threshold of 1%. A step size of 0.5 mm was used for the human brain, and a step size of 0.2 mm was used for the macaque brain. Resultant spatial path distributions were normalised by the total number of valid streamlines.

### Registration to standard space

For the human data, nonlinear transformations of T1-weighted (T1w) to MNI152 standard space were obtained. The distortion-corrected dMRI data were separately linearly aligned to the T1w space, and the concatenation of the diffusion-to-T1w and T1w-to-MNI transforms allowed diffusion-to-MNI warp fields to be obtained. For the macaque, nonlinear transformations to the macaque F99 standard space were estimated using FSL's FNIRT (*Andersson et al., 2007*) based on the corresponding FA maps (Key resources table). For cases where NMT-space tractography protocols were used, nonlinear transformations to NMT space were obtained using RheMAP (*Klink and Sirmpilatze, 2020*) (Key resources table).

## Tractography against data quality and individual variability

### Varying data quality

To explore robustness against varying data quality, we compared tractography reconstructions for in vivo human dMRI data of considerably different data resolutions, diffusion contrast, and scan time. Specifically, we explored whether tract reconstructions in state-of-the-art HCP data (approximately 55 min of scan time) were similar to reconstructions in bog standard data from the UK Biobank (UKB) (approximately 6.5 min of scan time), both on group-average maps, as well as individual reconstructions.

Inter-subject variability for each tract reconstruction was assessed within and across the HCP and UKB cohorts. Inter-subject Pearson's correlations were obtained by cross-correlating random subject pairs tract-wise. Specifically, for each subject pair, we correlated the normalised path distributions in MNI space for each tract, after thresholding the path distribution at 0.5% (*Warrington et al., 2020*), and then averaged the correlation across tracts for each subject pair. This was repeated for all possible unique subject pairs within and across cohorts.

A pairwise Mann–Whitney *U*-test was performed to determine differences in variability across analyses. For example, we compared the HCP vs UKB correlation between original and the new (+revised) tracts. We corrected for multiple comparisons using Bonferroni correction.

We also explored tract reconstructions on individual subjects. To demonstrate representative results, we ranked subjects based on their tractography results against the cohort average and picked the 10th, 50th (median), and 90th percentiles of the subjects. Specifically, for each subject, we calculated the average Pearson's correlation value, to the group average, across all tracts. We then ranked the subjects based on this value.

### Respecting similarities stemming from twinship

As an indirect way to explore whether the proposed standardised protocols respected individual variability, we tested whether tractography reconstructions reflected similarities stemming from twinship. We used the family structure in the HCP cohort to explore whether tracts of monozygotic twin pairs were more similar compared to tract similarity in dizygotic twins and non-twin sibling pairs, and to tract similarity in unrelated subject pairs, as would be expected by heritability of structural connections (*Bohlken et al., 2014*; *Jansen et al., 2015*; *Shen et al., 2014*). We used the 72 pairs of monozygotic twins (MZ) available in the HCP cohort, and randomly selected 72 pairs of dizygotic twins (DZ), 72 pairs

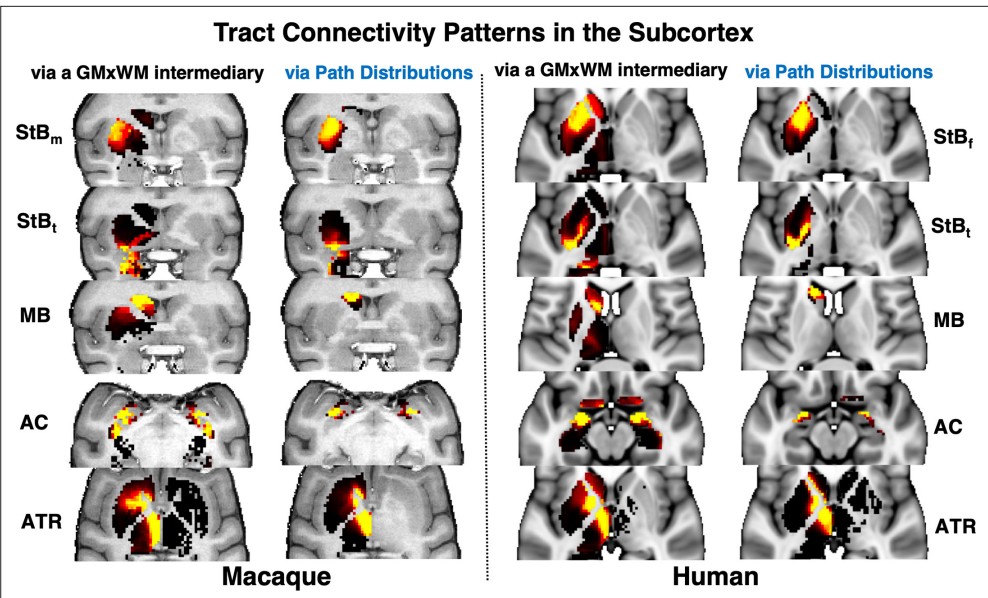

**Figure 9.** Improved specificity in subcortical connectivity patterns when using directly the tractography path distributions. Subcortical $GM_{sub} \times Tracts$ blueprints were built using: (1) an intermediary whole-brain tractography $GM \times WM$ matrix, multiplied by $WM \times Tracts$ as done in **Mars et al., 2018b** for cortical regions, and (2) the intersection of the path distribution of each tract with the subcortical structures of interest. The two approaches are shown on the left and right columns for each of the macaque and human examples and for representative example tracts (rows). The latter approach resulted in improved specificity in both the macaque and human, with the tract of interest connecting more focally to the relevant subcortical nucleus. For instance *StB* tracts end up more specifically in the putamen, *MB* in the caudate, *AC* in the amygdala, and *ATR* in the thalamus. All examples are shown as axial views, apart from $StB_m$, $StB_t$, MB in the macaque that are shown in coronal views.

of non-twin siblings, and 72 pairs of unrelated subjects, to have a balanced comparison. We compared tracts across pairs to assess whether our automated protocols respect the underlying tract variability across individuals. Specifically, for a given subject pair and a given tract, we calculated the Pearson's correlation between the normalised path distributions (in MNI space and following thresholding at 0.5%). We repeated this for all tracts and then calculated the mean correlation and standard deviation across tracts for that subject pair. This was then repeated for each group of subject pairs, giving a distribution of average correlations for each group. We subsequently compared these distributions between the different groups. We repeated this process separately for the Original XTRACT tracts (**Warrington et al., 2020**) and the new cortico-subcortical tracts to ensure that patterns were similar. For each analysis, a pairwise Mann–Whitney *U*-test was performed for all cohort pairs to determine the significant differences between them. We corrected for multiple comparisons using Bonferroni.

### Building connectivity blueprints in cortex and subcortex

Connectivity blueprints are $GM \times Tracts$ matrices that have been proposed to represent the pattern of connections of GM areas to a predefined set of WM tracts (**Mars et al., 2018b**; **Mars et al., 2021**). To do so, the intersection of the core of WM tracts with the WM–GM boundary needs to be identified. For cortical GM, simply obtaining the intersection from the spatial path distribution maps of each tract would be dominated by the gyral bias in tractography near the cortex (**Van Essen, 2014**). Instead, whole-brain tractography matrices can be used as intermediaries. Specifically a $GM \times WM$ connectivity matrix can be generated by seeding from each location of the WM–GM boundary and targeting to a whole WM mask and this can then be multiplied by a $WM \times Tracts$ obtained by collating the path distributions of all tracts of interest.

The cortical blueprints $GM_{ctx} \times Tracts$ were generated using our previously developed tool `xtract_blueprint` (**Mars et al., 2018b**; **Warrington et al., 2022**). We used the GM–WM boundary surface, extracted using the HCP pipelines (**Glasser et al., 2013**) for the human data and the approach in **Mars et al., 2018b** for the macaque. Briefly, a single set of macaque surfaces was derived using a set

of high-quality structural data from one of the macaque subjects. The remaining macaque data were then nonlinearly transformed to this space, and the surfaces were nonlinearly transformed to the F99 standard space. All surface data were downsampled to 10,000 vertices prior to tractography. Volume space WM targets were downsampled to 3 mm isotropic for the human and 2 mm isotropic for the macaque.

We extended the blueprint generation to include the subcortex. For subcortical nuclei, we found that using an intermediary $GM \times WM$ matrix did not help (as gyral bias is not relevant in subcortex – in fact, it made patterns less specific). Hence, subcortical $GM_{sub} \times Tracts$ blueprints were built using the intersection of the path distribution of each tract with the subcortical structures of interest (i.e. through multiplication of WM tracts and binary subcortical masks, including putamen, caudate, thalamus, hippocampus, and amygdala). *Figure 9* shows a comparison of the two approaches for various tracts in the human and macaque: (1) using an intermediary $GM \times WM$ matrix to obtain subcortical connection patterns, as done in *Mars et al., 2018b* for cortical regions and (2) using directly the tractography path distributions. The latter approach gave more focal and specific patterns and was used here for the subcortical regions. Tracts were downsampled (at 2 mm for human and 1 mm for macaque), thresholded at 0.1%, and multiplied by the subcortical nuclei masks, and then vectorised and stacked to create a $GM_{sub} \times Tracts$ matrix. These were then row-wise concatenated (i.e vertically) with the cortical blueprints to generate CIFTI-style blueprints with approximately 10,000 cortical vertices and approximately 5000 subcortical voxels (per left/right hemisphere). Finally, connectivity blueprints were row-wise sum-normalised. Following subject-wise construction of connectivity blueprints, we derived group-averaged blueprints for macaques and humans.

## Comparing connectivity blueprints across species

We compared GM connectivity patterns between humans and macaques (i.e. rows of the corresponding connectivity blueprint matrices), both in cortex and subcortex. As connectivity patterns are anchored by sets of homologously defined WM landmarks, connectivity patterns may be compared statistically using Kullback–Leibler (KL) divergence (*Equation 1*; *Kullback and Leibler, 1951*), as previously used (*Mars et al., 2018b*).

Let M be the macaque connectivity blueprint matrix, with $M_{ik}$ linking GM (cortex or subcortex) location $i$ to tract $k = 1 : T$, with the set of tracts with length T. Let matrix H be the equivalent matrix for the human brain. Vertices $i$ and $j$ in the macaque and human brains can then be compared in terms of their connectivity patterns $M_{ik}$, $H_{jk}$, $k = 1 : T$ using the symmetric KL divergence $D_{ij}$ as a dissimilarity measure. To avoid degeneracies in KL divergence calculations induced by the presence of zeros, we shifted all blueprint values by $\delta = 10^{-6}$. We used the tool `xtract_divergence` to perform all relevant calculations.

$$D_{ij} = \sum_k M_{ik} \log_2 \frac{M_{ik}}{H_{jk}} + \sum_k H_{jk} \log_2 \frac{H_{jk}}{M_{ik}}. \tag{1}$$

## Acknowledgements

Funding: SA, SW, and SNS acknowledge funding from the European Research Council (ERC Consolidator – 101000969 to SNS). SNS, SJ, and SH acknowledge support from the Centre for Mesoscale Connectomics (NIH UM1NS132207). The Centre for Integrative Neuroimaging was supported by the Wellcome Trust [203139/Z/16/Z]. RBM acknowledges funding from the Medical Research Council UK [MR/Y010698/1].

## Additional information

### Competing interests

Saad Jbabdi: Reviewing editor, *eLife*. The other authors declare that no competing interests exist.

## Funding

| Funder | Grant reference number | Author |
|---|---|---|
| European Research Council | ERC 101000969 | Stephania Assimopoulos<br>Shaun Warrington<br>Stamatios N Sotiropoulos |
| National Institutes of Health | NIH UM1NS132207 | Saad Jbabdi<br>Sarah Heilbronner<br>Stamatios N Sotiropoulos |
| Wellcome Trust | 10.35802/203139 | Saad Jbabdi<br>Rogier B Mars |
| Medical Research Council | MR/Y010698/1 | Rogier B Mars |

The funders had no role in study design, data collection, and interpretation, or the decision to submit the work for publication. For the purpose of Open Access, the authors have applied a CC BY public copyright license to any Author Accepted Manuscript version arising from this submission.

## Author contributions

Stephania Assimopoulos, Conceptualization, Data curation, Software, Formal analysis, Validation, Investigation, Visualization, Methodology, Writing – original draft, Writing – review and editing; Shaun Warrington, Data curation, Software, Investigation, Methodology, Writing – review and editing; Davide Folloni, Rogier B Mars, Formal analysis, Methodology, Writing – review and editing; Katherine Bryant, Wei Tang, Saad Jbabdi, Sarah R Heilbronner, Resources, Methodology, Writing – review and editing; Ali-Reza Mohammadi-Nejad, Resources, Methodology; Stamatios N Sotiropoulos, Conceptualization, Resources, Formal analysis, Supervision, Funding acquisition, Methodology, Writing – original draft, Writing – review and editing

## Author ORCIDs

Stephania Assimopoulos ⬤ https://orcid.org/0000-0002-7321-6297
Shaun Warrington ⬤ https://orcid.org/0000-0002-7198-8162
Katherine Bryant ⬤ http://orcid.org/0000-0003-1045-4543
Ali-Reza Mohammadi-Nejad ⬤ http://orcid.org/0000-0002-0710-6700
Saad Jbabdi ⬤ https://orcid.org/0000-0003-3234-5639
Sarah R Heilbronner ⬤ https://orcid.org/0000-0003-0893-5364
Rogier B Mars ⬤ https://orcid.org/0000-0001-6302-8631
Stamatios N Sotiropoulos ⬤ https://orcid.org/0000-0003-4735-5776

Reviewer #1 (Public review): https://doi.org/10.7554/eLife.107012.3.sa1
Reviewer #2 (Public review): https://doi.org/10.7554/eLife.107012.3.sa2
Author response https://doi.org/10.7554/eLife.107012.3.sa3

# Additional files

## Supplementary files

MDAR checklist

## Data availability

Human in vivo diffusion MRI data are publicly available (https://www.humanconnectome.org) and provided by the Human Connectome Project (HCP; https://www.humanconnectome.org), WU-Minn Consortium (Principal Investigators: David van Essen and Kamil Ugurbil; 1U54MH091657) funded by the 16 NIH Institutes and Centres that support the NIH Blueprint for Neuroscience Research; and by the McDonnell Centre for Systems Neuroscience at Washington University (*Van Essen et al., 2013*). The UK Biobank (https://www.ukbiobank.ac.uk) data were used under UK Biobank Project 43822 (PI: Sotiropoulos). Macaque data are openly available (https://fcon_1000.projects.nitrc.org/indi/PRIME/oxford2.html) and provided via the PRIMatE Data Exchange (http://fcon_1000.projects.nitrc.org/indi/PRIME/oxford2.html) (*Milham et al., 2018*). Tractography protocols (https://github.com/SPMIC-UoN/xtract_data copy archived at *Warrington, 2024*) and white matter tract atlases (https://github.com/

SPMIC-UoN/XTRACT_atlases copy archived at *Warrington, 2021*) are made available on GitHub. Tools for performing standardised and automated tractography (XTRACT), building connectivity blueprints (xtract blueprint), and performing divergence-based comparisons of connectivity blueprints (xtract divergence) are available on GitHub (https://github.com/SPMIC-UoN/xtract) and are released in FSL (v6.0.7.10 onwards, https://fsl.fmrib.ox.ac.uk/fsl/docs/#/diffusion/xtract). The cortico-subcortical protocols are available on GitHub (https://github.com/SPMIC-UoN/XTRACT_subcortex copy archived at *Assimopoulos and Sotiropoulos, 2025*).

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

# Appendix 1

## Original cortico-thalamic XTRACT protocols

For completeness of presenting all the cortico-subcortical protocols, we include here the previously published protocols of thalamic radiations (*Warrington et al., 2020*)

### Acoustic radiation ($AR$)

The acoustic radiation connects the medial geniculate nucleus ($MGN$) of the thalamus to the auditory cortex. The seed was placed in the transverse temporal gyrus and the target covered the $MGN$ of the thalamus. The exclusion mask consisted of two coronal planes, anterior and posterior to the thalamus, and an axial plane superior to the thalamus. In addition, the exclusion mask contained the brainstem and a horizontal region covering the optic tract.

### Anterior thalamic radiation ($ATR$)

The anterior thalamic radiation connects the thalamus to the frontal lobe. The seed mask was a coronal mask through the anterior part of the thalamus (120), with a coronal target mask at the anterior thalamic peduncle. In addition, the exclusion mask contained an axial plane covering the base of the midbrain, a coronal plane preventing leakage via the posterior thalamic peduncle and a coronal plane preventing leakage via the cingulum. A coronal stop mask covered the posterior part of the thalamus, extending from the base of the midbrain to the callosal sulcus.

### Optic radiation ($OR$)

The optic radiation consists of fibres from the lateral geniculate nucleus ($LGN$) of the thalamus to the primary visual cortex. The seed was placed in the $LGN$ and the target mask consisted of a coronal plane through the anterior part of the calcarine fissure. Exclusion masks consisted of an axial block of the brainstem, a coronal block of fibres directly posterior to the $LGN$ to select fibres that curl around dorsally, and a coronal plane anterior to the seed to prevent leaking into longitudinal fibres.

### Superior thalamic radiation ($STR$)

The superior thalamic radiation connects the thalamus to the pre-/post-central gyrus, respectively. The seed was a mask covering the whole thalamus and the target an axial plane covering the superior thalamic peduncle. An axial plane was used as a stop mask ventrally to the thalamus. The exclusion mask included two coronal planes, anterior and posterior to the target, to exclude tracking to the prefrontal cortex and occipital cortex, respectively.

**Appendix 1—table 1.** List of all species-matched (human and macaque) tract protocols, grouped as cortico-cortical, cortico-subcortical, and cerebellar.
Columns indicate whether corresponding protocols are bilateral or not and whether they are New, Revised, or not changed (Original XTRACT).

| Cortico-cortical | Abbreviation | Bilateral | Version |
|---|---|---|---|
| Arcuate fasciculus | *AF* | Yes | Original |
| Cingulum subsection: dorsal | *CBD* | Yes | Original |
| Cingulum subsection: peri-genual | *CBP* | Yes | Original |
| Cingulum subsection: temporal | *CBT* | Yes | Original |
| Corticospinal tract | *CST* | Yes | Original |
| Frontal aslant | *FA* | Yes | Original |
| Forceps major | *FMA* | No | Original |
| Forceps minor | *FMI* | No | Original |
| Inferior longitudinal fasciculus | *ILF* | Yes | Original |

*Appendix 1—table 1 Continued on next page*

*Appendix 1—table 1 Continued*

| Cortico-cortical | Abbreviation | Bilateral | Version |
|---|---|---|---|
| Inferior fronto-occipital fasciculus | *IFO* | Yes | Original |
| Middle longitudinal fasciculus | *MdLF* | Yes | Original |
| Superior longitudinal fasciculus 1 | *SLF*1 | Yes | Original |
| Superior longitudinal fasciculus 2 | *SLF*2 | Yes | Original |
| Superior longitudinal fasciculus 3 | *SLF*3 | Yes | Original |
| Vertical occipital fasciculus | *VOF* | Yes | Original |
| Uncinate fasciculus | *UF* | Yes | Revised |
| Cortico-subcortical | | | |
| Acoustic radiation | *AR* | Yes | Original |
| Anterior thalamic radiation | *ATR* | Yes | Original |
| Optic radiation | *OR* | Yes | Original |
| Superior thalamic radiation | *STR* | Yes | Original |
| Fornix | *FX* | Yes | Revised |
| Anterior commissure | *AC* | No | Revised |
| Amygdalofugal tract | *AMF* | Yes | New |
| Muratoff bundle/subcallosal fasciculus | *MB* | Yes | New |
| Striatal bundle/external capsule (sensorimotor) | $StB_m$ | Yes | New |
| Striatal bundle/external capsule (frontal) | $StB_f$ | Yes | New |
| Striatal bundle/external capsule (temporal) | $StB_t$ | Yes | New |
| Striatal bundle/external capsule (parietal) | $StB_p$ | Yes | New |
| Extreme capsule (frontal) | $EmC_f$ | Yes | New |
| Extreme capsule (temporal) | $EmC_t$ | Yes | New |
| Extreme capsule (parietal) | $EmC_p$ | Yes | New |
| Cerebellar | | | |
| Middle cerebellar peduncle | MCP | No | Original |

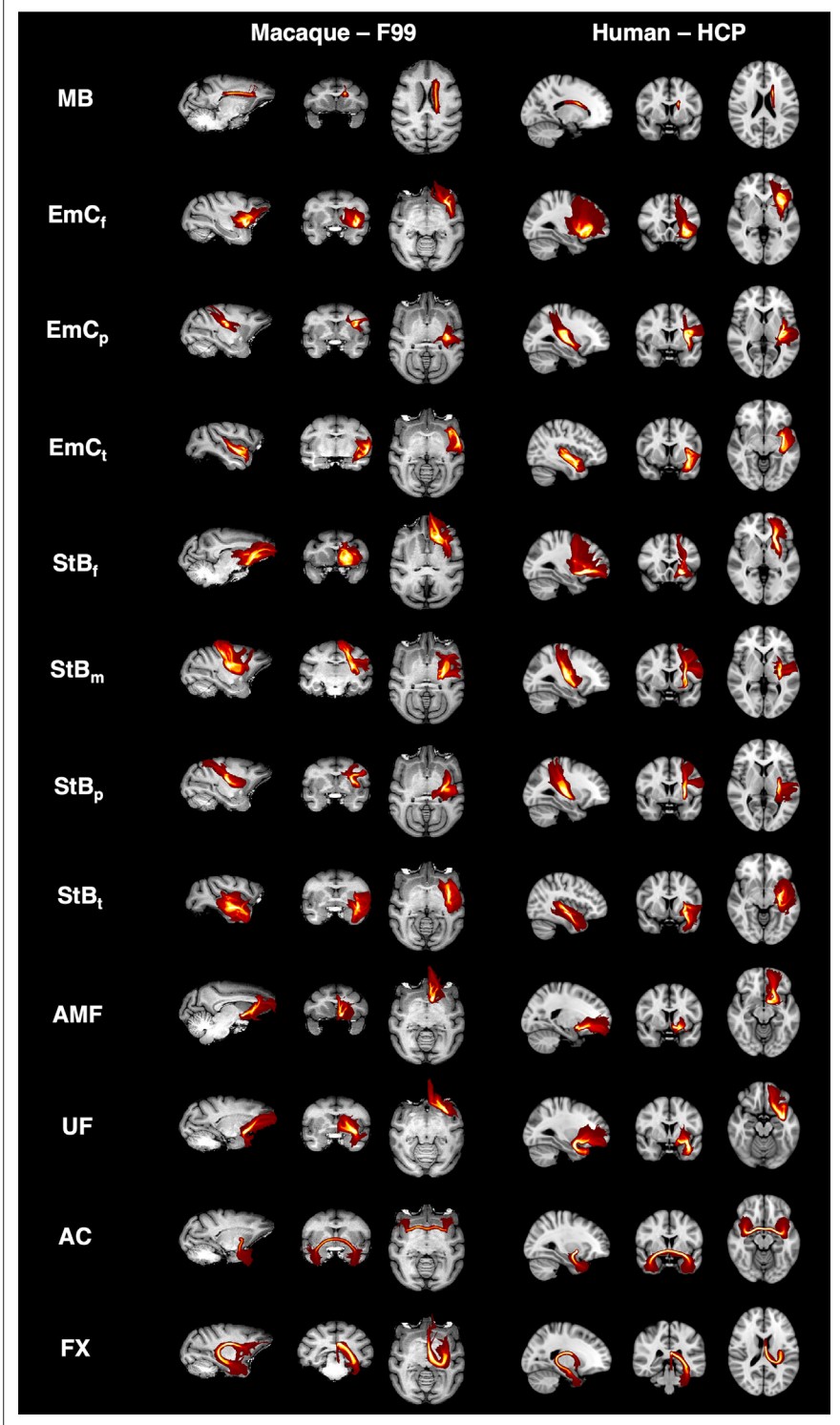

**Appendix 1—figure 1.** Tract reconstructions using our cortico-subcortical protocols, with good agreement between the macaque and the human. Maximum intensity projections (MIPs) of the group-averaged path distributions for all developed tractography protocols in the macaque (6 animal average) and human (50 healthy subject average from the Human Connectome Project dataset; HCP). All MIPs are across a window (20% of the field of view) centred at the displayed slices. Thresholded path distributions are displayed with a low threshold of 0.1% (for the *EmC* parts the 90th percentile was used).

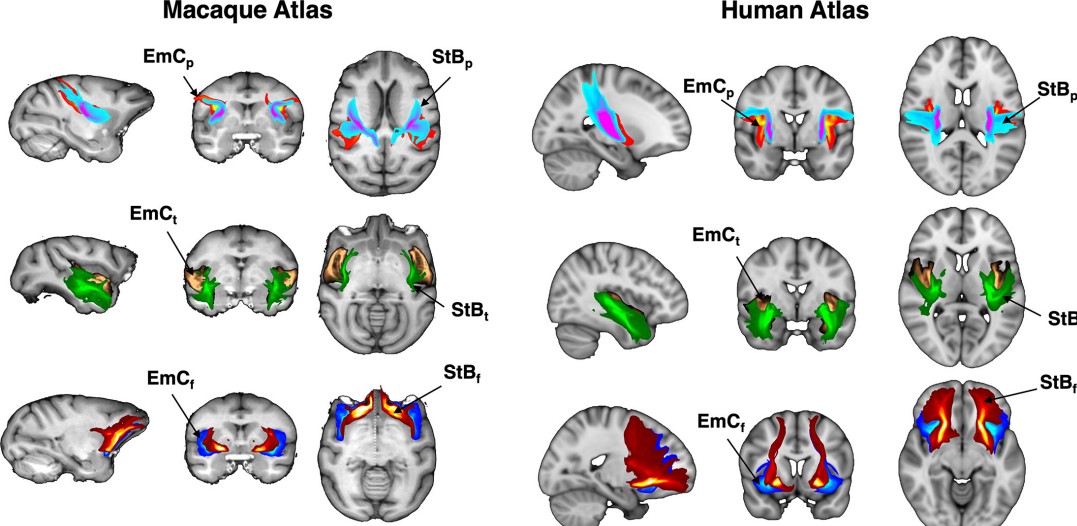

**Appendix 1—figure 2.** Relative positions maintained for all corresponding parts of striatal and extreme capsule bundles, across species. Maximum intensity projections (MIPs) of the group-averaged tractography results for corresponding parts of the striatal bundle (*StB*)/external capsule and the extreme capsule (*EmC*) in the macaque (6 animal average) and human (50 healthy subject average from the Human Connectome Project dataset; HCP). For each part, *StB* is more medial and *EmC* is more lateral with respect to each other. Tracts considered: frontal, temporal, and parietal parts of the anterior limb of the extreme capsule (*EmC_f* , *EmC_t* , *EmC_p*); frontal, temporal, and parietal parts of the striatal bundle (*StB_f* ,*StB_t* , *StB_p*) (**Table 1** in main text).

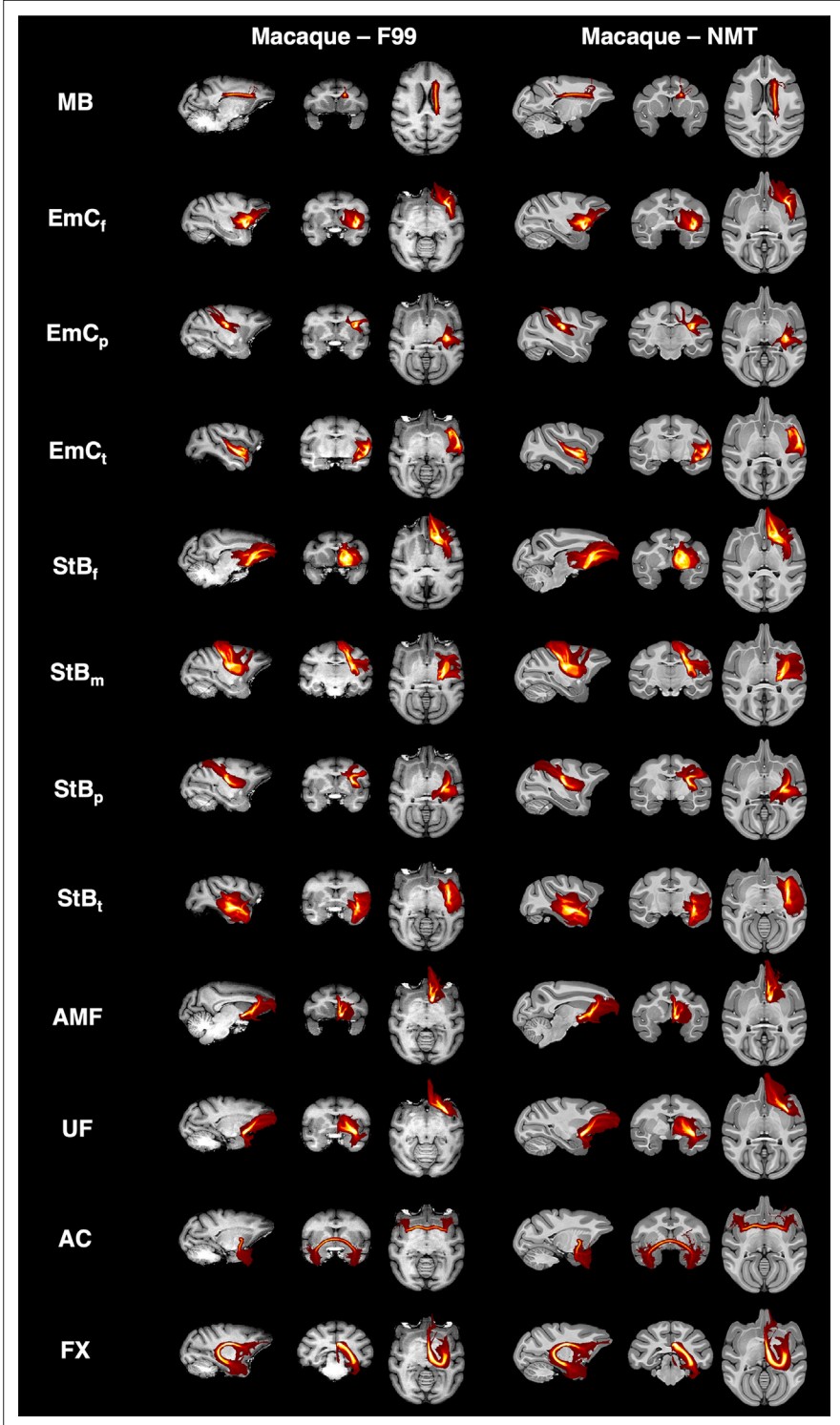

**Appendix 1—figure 3.** Corresponding tract reconstructions using our macaque cortico-subcortical protocols, between the two macaque standard spaces, F99 and NMT. Maximum intensity projections (MIPs) of the group-averaged path distributions for all developed tractography results for all developed protocols in the macaque (6 animal average) using protocols in the F99 standard space and protocols in the NMT standard space. All MIPs are across a window (20% of the field of view) centred at the displayed slices. Thresholded path distributions are displayed with a low threshold of 0.1% (for the *EmC* parts the 90th percentile was used).

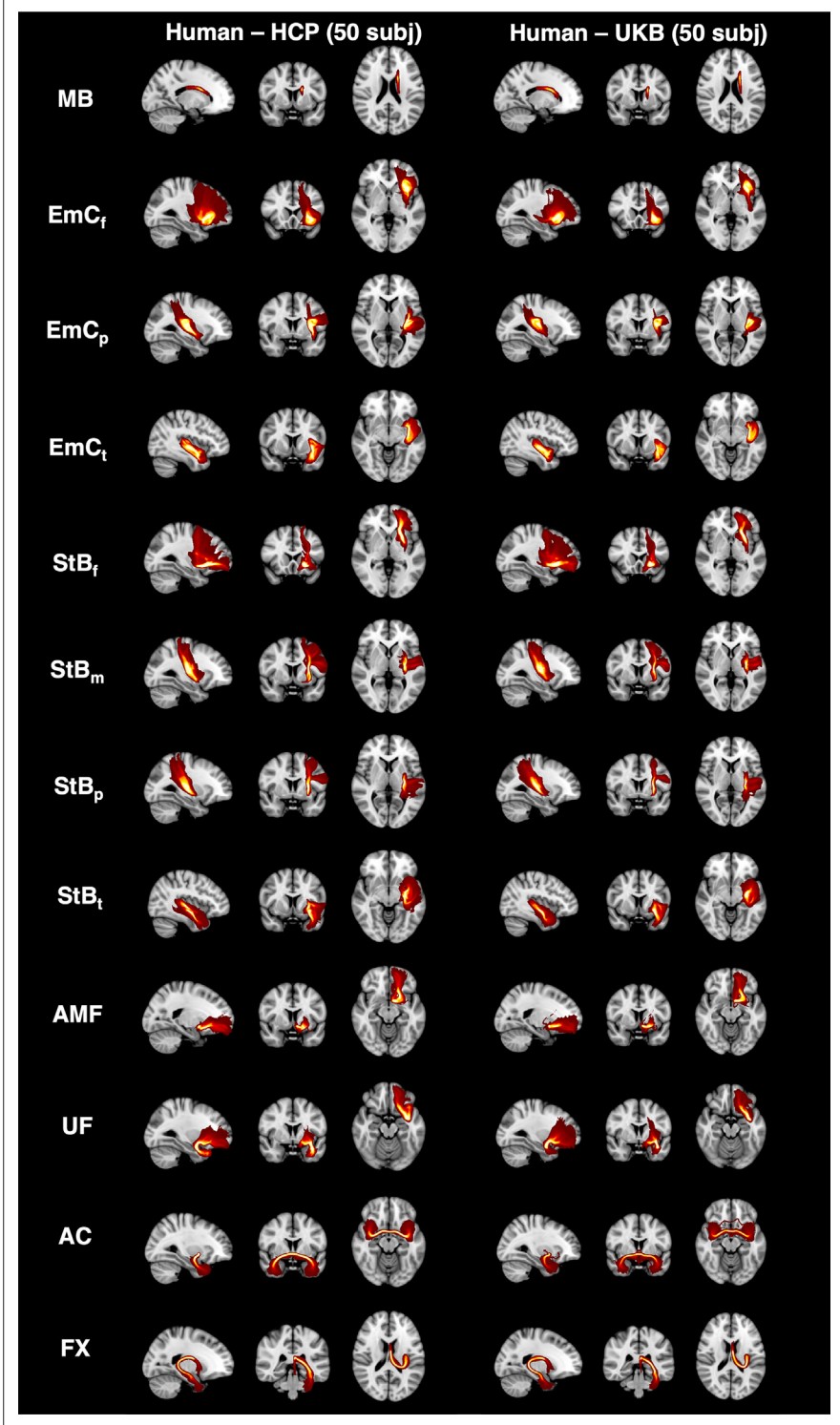

**Appendix 1—figure 4.** Corresponding tract reconstructions using our cortico-subcortical protocols, between different data resolutions (spatial and angular) and acquisition protocols. Maximum intensity projections (MIPs) of the group-averaged path distributions for all developed tractography results for all developed protocols in 50 Human Connectome Project (HCP) and 50 UK Biobank (UKB) subjects. All MIPs are across a window (20% of the field of view) centred at the displayed slices. Thresholded path distributions are displayed with a low threshold of 0.1% (for the *EmC* parts the 90th percentile was used).

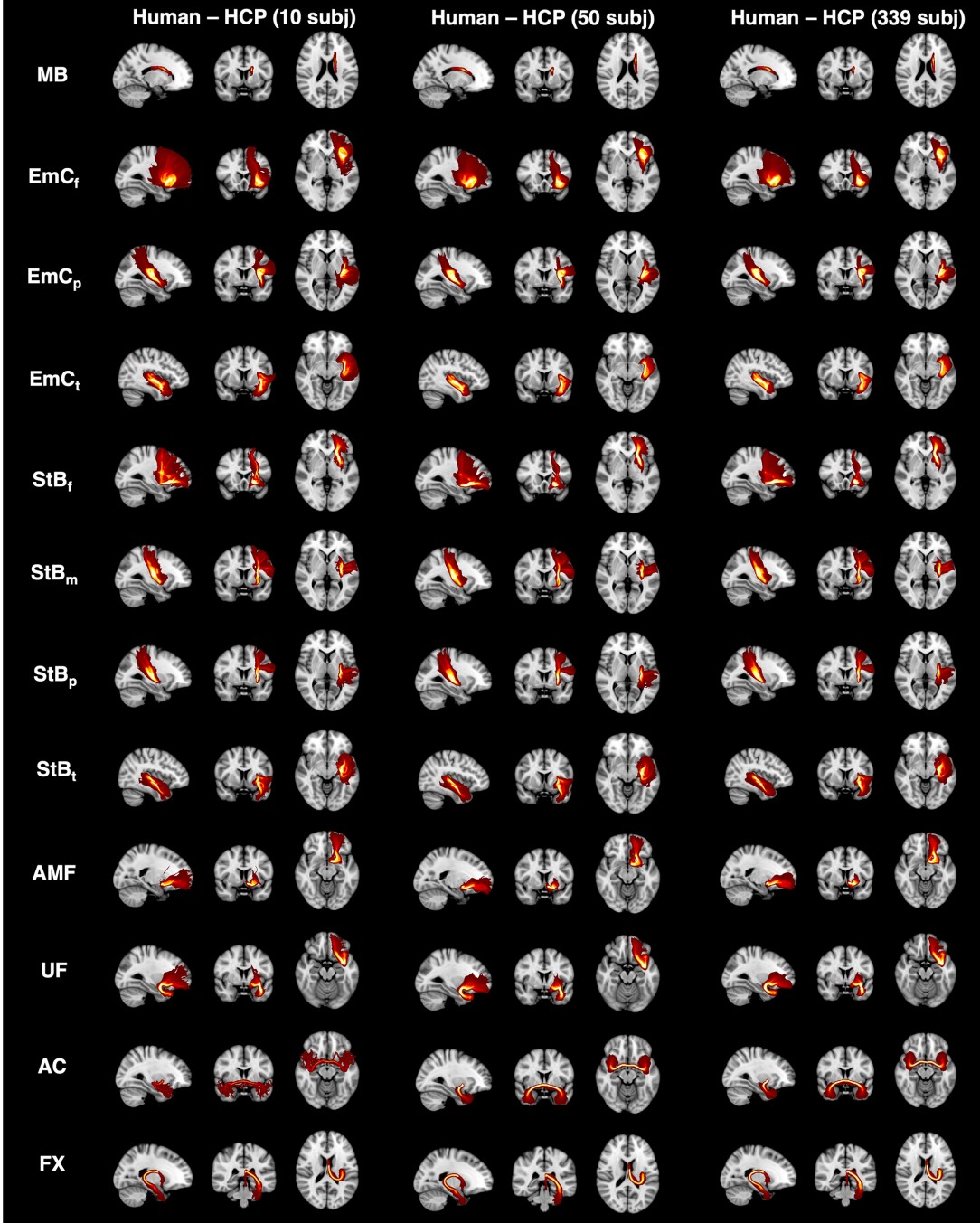

**Appendix 1—figure 5.** Corresponding tract reconstructions using our cortico-subcortical protocols, across different cohort sizes. Maximum intensity projections (MIPs) of the group-averaged path distributions for all developed tractography results for all developed protocols in 10, 50, and 339 (all unrelated subjects) Human Connectome Project (HCP) subjects. All MIPs are across a window (20% of the field of view) centred at the displayed slices. Thresholded path distributions are displayed with a low threshold of 0.1% (for the *EmC* parts the 90th percentile was used).

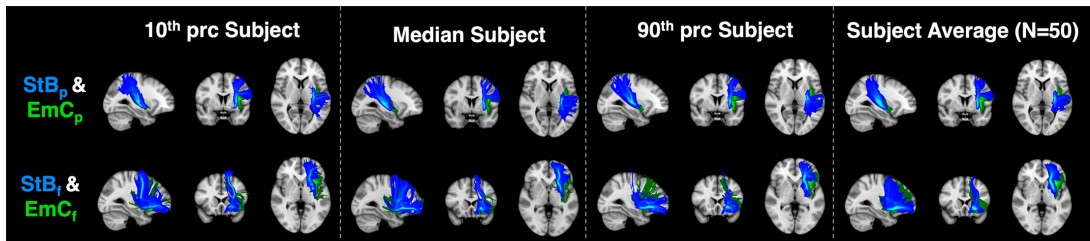

**Appendix 1—figure 6.** Individual subject tract reconstructions match the group average, while preserving expected topology for *StB* and *EmC*. Showing reconstruction and relative topology for frontal and parietal parts ($StB_f$ vs $EmC_f$, $StB_p$ vs $EmC_p$) for individual subjects. The subjects chosen are those corresponding to the 10th, 50th (median), and 90th percentile of the distribution of tract correlations against the group average for the HCP cohort ($N$ = 50). For each subject, the mean correlation to the average across all New tracts was computed and subjects were ranked based on this mean tract correlation. In all cases, tracts reconstruct similarly to the average atlas, while preserving the expected topology (*StB* more medial than *EmC*).

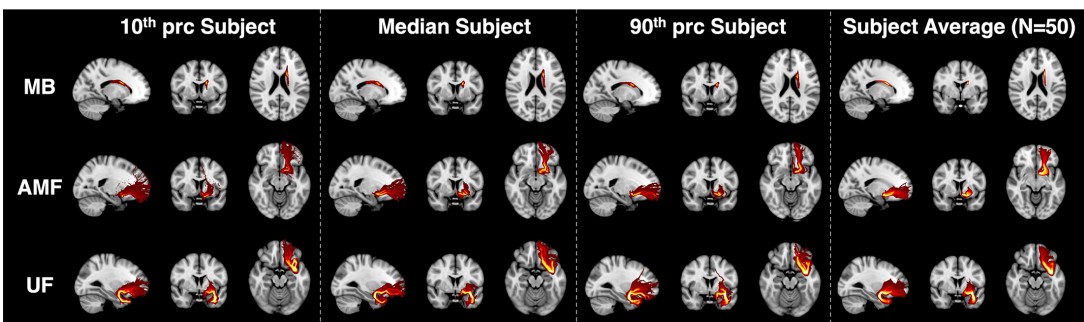

**Appendix 1—figure 7.** Individual subject tract reconstructions match the group average for *MB*, *AMF*, and *UF*. The subjects chosen are those corresponding to the 10th, 50th (median), and 90th percentiles of the tract correlations against the group average for the HCP cohort ($N$ = 50). For each subject, the mean correlation to the average across all New tracts was computed and subjects were ranked based on this mean tract correlation. In all cases, tracts reconstruct similarly to the atlas.

