## [Editor Report · eLife Assessment]

This **important** study provides a novel approach for delineating subcortical-cortical white matter bundles. The authors provide **convincing** evidence by harnessing state-of-the-art methods and cross-species data. Together, this effort will be of interest to scientists across multiple subfields and accelerate progress in a biologically critical but methodologically challenging area.

---

## [Referee Report · Reviewer #1 (Public review)]

The authors note that it is challenging to perform diffusion MRI tractography consistently in both humans and macaques, particularly when deep subcortical structures are involved. The scientific advance described in this paper is effectively an update to the tracts that the XTRACT software supports. The changes to XTRACT are soundly motivated in theory (based on anatomical tracer studies) and practice (changes in seeding/masking for tractography).

---

## [Referee Report · Reviewer #2 (Public review)]

Summary:

In this article, Assimopoulos et al. expand the FSL-XTRACT software to include new protocols for identifying cortical-subcortical tracts with diffusion MRI, with a focus on tracts connecting to the amygdala and striatum. They show that the amygdalofugal pathway and divisions of the striatal bundle/external capsule can be successfully reconstructed in both macaques and humans while preserving large-scale topographic features previously defined in tract tracing studies. The authors set out to create an automated subcortical tractography protocol, and they accomplish this for a subset of specific subcortical connections.

Strengths:

The main strength of the current study is the translation of established anatomical knowledge to a tractography protocol for delineating cortical-subcortical tracts that are difficult to reconstruct. Diffusion MRI-based tractography is highly prone to false positives; thus, constraining tractography outputs by known anatomical priors is important. The authors used existing tracing literature to create anatomical constraints for tracking specific cortical-subcortical connections and refined their protocol through an iterative process and in collaboration with multiple neuroanatomists. Key additional strengths include (1) the creation of a protocol that can be applied to both macaque and human data; (2) demonstration that the protocol can be applied to be high quality data (3 shells, > 250 directions, 1.25 mm isotropic, 55 minutes) and lower quality data (2 shells, 100 directions, 2 mm isotropic, 6.5 minutes); and (3) validation that the anatomy of cortical-subcortical tracts derived from the new method are more similar in monozygotic twins than in siblings and unrelated individuals.

Overall Appraisal:

This new method will accelerate research on anatomically validated cortical-subcortical white matter pathways. The work has utility for diffusion MRI researchers across fields.

Editors' note:

Both reviewers were satisfied with the responses to their feedback.

---

## [Author Response]

The following is the authors’ response to the original reviews.

**Reviewer #1 (Public Review):**
(1) Summary:The authors note that it is challenging to perform diffusion MRI tractography consistently in both humans and macaques, particularly when deep subcortical structures are involved. The scientific advance described in this paper is effectively an update to the tracts that the XTRACT software supports. The claims of robustness are based on a very small selection of subjects from a very atypical dMRI acquisition (n=50 from HCP-Adult) and an even smaller selection of subjects from a more typical study (n=10 from ON-Harmony).Strengths:The changes to XTRACT are soundly motivated in theory (based on anatomical tracer studies) and practice (changes in seeding/masking for tractography), and I think the value added by these changes to XTRACT should be shared with the field. While other bundle segmentation software typically includes these types of changes in release notes, I think papers are more appropriate.

We would like to thank the reviewer for their assessment and we appreciate the comments for improving our manuscript. We have added new results, sampling from a larger cohort with a typical dMRI protocol (N=50 from UK Biobank), as well as showcasing examples from individual subject reconstructions (Supplementary figures S6, S7). We also demonstrate comparisons against another approach that has been proposed for extracting parts of the cortico-striatal bundle in a bundle segmentation fashion, as the reviewer suggests (see comment and Author response image 1 below).

We would also like to take the opportunity to summarise the novelty of our contribuIons, as detailed in the Introduction, which we believe extend beyond a mere software update; this is a byproduct of this work rather than the aim.

i) We devise for the first Ime standard-space protocols for 21 challenging cortico-subcortical bundles for both human and macaque and we interrogate them in a comprehensive manner.

ii) We demonstrate robustness of these protocols using criteria grounded on neuroanatomy, showing that tractography reconstructions follow topographical principles known from tracers both in WM and GM and for both species. We also show that these protocols capture individual variability as assessed by respecting family structure in data from the HCP twins.

iii) We use high-resolution dMRI data (HCP and post-mortem macaque) to showcase feasibility of these reconstructions, and we show that reconstructions are also plausible with more conventional data, such as the ones from the UK Biobank.

iv) We further showcase robustness and the value of cross-species mapping by using these tractography reconstructions to predict known homologous grey matter (GM) regions across the two species, both in cortex and subcortex, on the basis of similarity of grey matter areal connection patterns to the set of proposed white matter bundles.

Weaknesses(2) The demonstration of the new tracts does not include a large number of carefully selected scans and is only compared to the prior methods in XTRACT. The small n and limited statistical comparisons are insufficient to claim that they are better than an alternative. Qualitatively, this method looks sound.

We appreciate the suggestion for larger sample size, so we performed the same analysis using 50 randomly drawn UK Biobank subjects, instead of ON-Harmony, matching the N=50 randomly drawn HCP subjects (detailed explanation in the comment below, Main text Figure 4A; Supplementary Figures S4). We also generated results using the full set of N=339 HCP unrelated subjects (Supplementary Figure S5 compares 10, 50 and 339 unrelated HCP subjects). We provide further details in the relevant point (3) below.

With regards to comparisons to other methods, there are not really many analogous approaches that we can compare against. In our knowledge there are no previous cross-species, standard space tractography protocols for the tracts we considered in this study (including Muratoff, amygdalofugal, different parts of extreme an external capsules, along with their neighbouring tracts). We therefore (i) directly compared against independent neuroanatomical knowledge and patterns (Figures 2, 3, 5), (ii) confirmed that patterns against data quality and individual variability that the new tracts demonstrate are similar to patterns observed for the more established cortical tracts (Figure 4), (iii) indirectly assessed efficacy by performing a demanding task, such as homologue identification on the basis of the tracts we reconstruct (Figures 6, 7).

We need to point out that our approach is not “bundle segmentation”, in the sense of “datadriven” approaches that cluster streamlines into bundles following full-brain tractography. The latter is different in spirit and assigns a label to each generated streamline; as full-brain tractography is challenging (Maier-Hein, Nature Comms 2017), we follow instead the approach of imposing anatomical constraints to miIgate for some of these challenges as suggested in (MaierHein, 2017).

Nevertheless, we used TractSeg (one of the few alternatives that considers corticostriatal bundles) to perform some comparisons. The Author response image below shows average path distributions across 10 HCP subjects for a few bundles that we also reconstruct in our paper (no temporal part of striatal bundle is generated by Tractseg). We can observe that the output for each tract is highly overlapping across subjects, indicating that there is not much individual variability captured. We also see the reduced specificity in the connectivity end-points of the bundles.

**Author response image 1. sa3fig1:** Comparison between 10-subject average for example subcortical tracts using TractSeg and XTRACT. We chose example bundles shared between our set and TractSeg. Per subject TractSeg produces a binary mask rather than a path distribution per tract. Furthermore, the mask is highly overlapping across subjects. Where direct correspondence was not possible, we found the closest matching tract. Specifically, we used ST_PREF for STBf, and merged ST_PREC with ST_POSTC to match StBm. There was no correspondence for the temporal part of StB.

We subsequently performed the twinness test using both TractSeg and XTRACT (Author response image 2), as a way to assess whether aspects of individual variability can be captured. Due to heritability of brain organisation features, we anticipate that monozygotic twins have more similar tract reconstructions compared to dizygoIc twins and subsequently non-twin siblings. This pattern is reproduced using our proposed approach, but not using TractSeg that provides a rather flat pattern.

**Author response image 2. sa3fig2:** Violin plots of the mean pairwise Pearson’s correlations across tracts between 72 monozygotic (MZ) twin pairs, 72 dizygotic (DZ) twin pairs, 72 non-twin sibling pairs, and 72 unrelated subject pairs from the Human Connectome Project, using Tractseg (left) and XTRACT (right). About 12 cortico-subcortical tracts were considered, as closely matched as possible between the two approaches. For Tractseg we considered: 'CA', 'FX', 'ST_FO', 'ST_M1S1' (merged ‘ST_PREC’ and ‘ST_POSTC’ to approximate the sensorimotor part of our striatal bundle), 'ST_OCC', 'ST_PAR', 'ST_PREF', 'ST_PREM', 'T_M1S1' (merged ‘T_PREC’ and ‘T_POSTC’ to approximate the sensorimotor part of our striatal bundle), 'T_PREF', 'T_PREM', 'UF'. For XTRACT we considered: 'ac', 'fx', 'StB_f_', 'StB_m_', 'StB_p_', 'StB_t_, 'EmC_f_', 'EmC_p_', 'EmC_t_', 'MB', 'amf', 'uf'. Showing the mean (μ) and standard deviation (σ) for each group. There were no significant di^erences between groups using TractSeg.

Taken together, these results indicate as a minimum that the different approaches have potentially different aims. Their different behaviour across the two approaches can be desirable and beneficial for different applications (for instance WM ROI segmentation vs connectivity analysis) but makes it challenging to perform like-to-like comparisons.

(3) “Subject selection at each stage is unclear in this manuscript. On page 5 the data are described as "Using dMRI data from the macaque (𝑁 = 6) and human brain (𝑁 = 50)". Were the 50 HCP subjects selected to cover a range of noise levels or subject head motion? Figure 4 describes 72 pairs for each of monozygotic, dizygotic, non-twin siblings, and unrelated pairs - are these treated separately? Similarly, NH had 10 subjects, but each was scanned 5 times. How was this represented in the sample construction?”

We appreciate the suggestions and we agree that some of the choices in terms of group sizes may have been confusing. Short answer is we did not perform any subject selection, subjects were randomly drawn from what we had available. The 72 twin pairs are simply the maximum number of monozygotic twin pairs available in the HCP cohort, so we used 72 pairs in all categories to match this number in these specific tests. The N=6 animals are good quality post-mortem dMRI data that have been acquired in the past and we cannot easily expand. For the rest of the points, we have now made the following changes:

We have replaced our comparison to the ON-Harmony dataset (10 subjects) with a comparison to 50 unrelated UK Biobank subjects (to match the 50 unrelated HCP subject cohort used throughout). Updated results can be seen in Figure 4A and Supplementary Figure S4. This allows a comparison of tractography reconstruction between high quality and more conventional quality data for the same N.

We looked at QC metrics to ensure our chosen cohorts were representaIve of the full cohorts we had available. The N=50 unrelated HCP cohort and N=50 unrelated UKBiobank cohorts we used in the study captured well the range of the full 339 unrelated HCP cohort and N=7192 UKBiobank cohort in terms of absolute/relative moion (Author response image 3A and 3B respectively). A similar pattern was observed in terms of SNR and CNR ranges Author response image 4.

We generated tractography reconstructions for single subjects, corresponding to the 10th percentile (P_10_), median and 90th percentile (P90) of the distributions with respect to similarity to the cohort average maps. These are now shown in Supplementary Figures S6, S7. We also checked the QC metrics for these single subjects and confirmed that average absolute subject moIon was highest for the P_10_, followed by the P_50_ and lowest for the P_90_ subject, capturing a range of within cohort data quality.

We generated reconstructions for an even larger HCP cohort (all 339 unrelated HCP subjects) and these look very similar to the N=50 reconstructions (Supplementary Figure S5).

**Author response image 3. sa3fig3:** Subsets chosen from the HCP and UKB reflect similar range of average motion (relative and absolute) to the corresponding full cohorts. (A) Absolute and relative motion comparison between N=50 and N=339 unrelated HCP subjects. (B) Absolute and relative motion comparison between N=50 and N=7192 super-healthy UKB subjects.

**Author response image 4. sa3fig4:** Average SNR and CNR values show similar range between the N=50 UKB subset and the full UK Biobank cohort of N=7192.

(4) In the paper, the authors state "the mean agreement between HCP and NH reconstructions was lower for the new tracts, compared to the original protocols (𝑝 < 10^−10). This was due to occasionally reconstructing a sparser path distribution, i.e., slightly higher false negative rate," - how can we know this is a false negative rate without knowing the ground truth?

We are sorry for the terminology, we have corrected this, as it was confusing. Indeed, we cannot call it false negaIve, what we meant is that reconstructions from lower resolution data for these bundles ended up being in general sparser than the ones from the high-resolution data, potentially missing parts of the tract. We have now revised the text accordingly.

**Reviewer #2 Public Review:**
(5) Summary:In this article, Assimopoulos et al. expand the FSL-XTRACT software to include new protocols for identifying cortical-subcortical tracts with diffusion MRI, with a focus on tracts connecting to the amygdala and striatum. They show that the amygdalofugal pathway and divisions of the striatal bundle/external capsule can be successfully reconstructed in both macaques and humans while preserving large-scale topographic features previously defined in tract tracing studies. The authors set out to create an automated subcortical tractography protocol, and they accomplished this for a subset of specific subcortical connections for users of the FSL ecosystem.Strengths:A main strength of the current study is the translation of established anatomical knowledge to a tractography protocol for delineating cortical-subcortical tracts that are difficult to reconstruct. Diffusion MRI-based tractography is highly prone to false positives; thus, constraining tractography outputs by known anatomical priors is important. Key additional strengths include (1) the creation of a protocol that can be applied to both macaque and human data; (2) demonstration that the protocol can be applied to be high quality data (3 shells, > 250 directions, 1.25 mm isotropic, 55 minutes) and lower quality data (2 shells, 100 directions, 2 mm isotropic, 6.5 minutes); and (3) validation that the anatomy of cortical-subcortical tracts derived from the new method are more similar in monozygotic twins than in siblings and unrelated individuals.

We thank the Reviewer for the globally posiIve evaluaIon of this work and the perInent comments that have helped us to improve the paper.

Weaknesses(6) Although this work validates the general organizational location and topographic organization of tractography-derived cortical-subcortical tracts against prior tract tracing studies (a clear strength), the validation is purely visual and thus only qualitative. Furthermore, it is difficult to assess how the current XTRACT method may compare to currently available tractography approaches to delineating similar cortical-subcortical connections. Finally, it appears that the cortical-subcortical tractography protocols developed here can only be used via FSL-XTRACT (yet not with other dMRI software), somewhat limiting the overall accessibility of the method.

We agree that a more quanItative comparison against gold standard tracing data would be ideal. However, there are practical challenges that prohibit such a comparison at this stage: (i) Access to data. There are no quantifiable, openly shared, large scale/whole brain tracing data available. The Markov study provided the only openly available weighted connectivity matrices measured by tracers in macaques (Markov, Cereb Cortex 2014), which are only cortico-cortical and do not provide the white matter routes, they only quantify the relative contrast in connection terminals. (ii) 2D microscopy vs 3D tractography. The vast majority of tracing data one can find in neuroanatomy labs is on 2D microscopy slices with restricted field of view, which is also the case for the data we had access to for this study. This complicates significantly like-to-like comparisons against 3D whole-brain tractography reconstructions. (iii) Quantifiability is even tricky in the case of gold standard axonal tracing, as it depends on nuisance factors, e.g. injection site, injection size, injection uniformity and coverage, which confound the gold-standard measurements, but are not relevant for tractography. For these reasons, a number of high-profile NIH BRAIN CONNECTS Centres (for instance https://connects.mgh.harvard.edu/, https://mesoscaleconnecIvity.org/) are resourced to address these challenges at scale in the coming years and provide the tools to the community to perform such quantitative comparisons in the future.

In terms of comparison with other approaches, we have performed new tests and detail a response to a similar comment (2) from Reviewer 1.

Finally, our protocols have been FSL-tested, but have nothing that is FSL specific. We cannot speak of performance when used with other tools, but there is nothing that prohibits translation of these standard space protocols to other tools. In fact, the whole idea behind XTRACT was to generate an approach open to external contributions for bundle-specific delineation protocols, both for humans and for non-human species. A number of XTRACT extensions that have been published over the last 5 years for other NHP species (Roumazeilles et al. (2020); Bryant et al. (2020); Wang et al. (2025)) and similar approaches have been used in commercial packages (Boshkovski et al, 2106, ISMRM 2022).

**Recommendations To the Authors:**
(7) Superiority of the FSL-XTRACT approach to delineating cortical-subcortical tracts. The Introduction of the article describes how "Tractography protocols for white matter bundles that reach deeper subcortical regions, for instance the striatum or the amygdala, are more difficult to standardize" due to the size, proximity, complexity, and bottlenecks associated with corticalsubcortical tracts. It would be helpful for the authors to better describe how the analytic approach adopted here overcomes these various challenges. What does the present approach do differently than prior efforts to examine cortical-subcortical connectivity?

There have not been many prior efforts to standardise cortico-subcortical connecIvity reconstructions, as we overview in the Introduction. As outlined in (Schilling et al. (2020), https://doi.org/10.1007/s00429-020-02129-z), tractography reconstructions can be highly accurate if we guide them using constraints that dictate where pathways are supposed to go and where they should not go. This is the philosophy behind XTRACT and all the proposed protocols, which provide neuroanatomical constraints across different bundles. At the same time these constraints are relatively coarse so that they are species-generalisable. We have clarified that in Discussion. The approach we took was to first identify anatomical constraints from neuroanatomy literature for each tract of interest independently, derive and test these protocols in the macaque, and then optimise in an iterative fashion until the protocols generalise well to humans and until, when considering groups of bundles, the generated reconstructions can follow topographical principles known from tract tracing literature. This process took years in order to perform these iterations as meticulously as we could. We have modified the first sections in Methods to reflect this better (3rd paragraph of 1st Methods section), as well as modified the third and second to last paragraphs of the Introduction (“We propose an approach that addresses these challenges…”).

(8) Relatedly, it is difficult to fully evaluate the utility of the current approach to dissecting cortical-subcortical tracts without a qualitative or quantitative comparison to approaches that already exist in the field. Can the authors show that (or clarify how) the FSL-XTRACT approach is similar to - or superior to - currently available methods for defining cortical-striatal and amygdalofugal tracts (e.g., methods they cite in the Introduction)?”

From the limited similar approaches that exist, we did perform some comparisons against TractSeg, please see Reply to Comment 2 from Reviewer 1. We have also expanded the relevant text in the introduction to clarify the differences:

“…However, these either uIlise labour-intensive single-subject protocols (22,26), are not designed to be generalisable across species (42, 43), or are based mostly on geometrically-driven parcellaIons that do not necessarily preserve topographical principles of connecIons (40). We propose an approach that addresses these challenges and is automated, standardised, generalisable across two species and includes a larger set of cortico-subcortical bundles than considered before, yielding tractography reconstructions that are driven by neuroanatomical constraints.”

(9) Future applications of the tractography protocol:It would be helpful for the authors to describe the contexts in which the automated tractography approach developed here can (and cannot) be applied in future studies. Are future applications limited to diffusion data that has been processed with FSL's BEDPOSTX and PROBTRACKX? Can FSL-XTRACT take in diffusion data modelled in other software (e.g., with CSD in mrtrix or with GQI in DSI Studio)? Can the seed/stop/target/exclusion ROIs be applied to whole-brain tractography generated in other software? Integration with other software suites would increase the accessibility of the new tract dissection protocols.

We have added some text in the Discussion to clarify this point. Our protocols have been FSLtested, but have nothing that is FSL specific. We cannot speak of performance of other tools, but there is nothing that prohibits translaIon of these standard space protocols to other tools. As described before, the protocols are recipes with anatomical constraints including regions the corresponding white matter pathways connect to and regions they do not, constructed with cross-species generalisability in mind. In fact a number of other packages (even commercial) have adopted the XTRACT protocols with success in the past, so we do not see anything in principle that prohibits these new protocols to be similarly adopted.

We cannot comment on the protocols’ relevance for segmenIng whole-brain tractograms, as these can induce more false posiIves than tractography reconstructions from smaller seed regions and may require stricter exclusions.

(10) It was great to see confirmation that the XTRACT approach can be successfully applied in both high-quality diffusion data from the HCP and in the ON-Harmony data. Given the somewhat degraded performance in the lower quality dataset (e.g., Figure 4A), can the authors speak to the minimum data requirements needed to dissect these new cortical-subcortical tracts? Will the approach work on single-shell, low b data? Is there a minimum voxel resolution needed? Which tracts are expected to perform best and worst in lower-quality data?

Thank you for these comments, even if we have not really tried in lower (spaIal and angular) resolution data, given the proximity of the tracts considered, as well as the small size of some bundles, we would not recommend lower resolution than those of the UK Biobank protocol. In general, we would consider the UK Biobank protocol (2mm, 2 shells) as the minimum and any modern clinical scanner can achieve this in 6-8 minutes. We hence evaluated performance from high quality HCP to lower quality UK Biobank data, covering a considerable range (scan Ime from 55 minutes down to 6 minutes).

In terms of which tract reconstructions were more reproducible for UKBiobank data, the tracts with lowest correlations across subjects (Figure 4) were the anterior commissure (AC) and the temporal part of the Extreme Capsule (EmC_t_), while the highest correlations were for the Muratoff Bundle (MB) and the temporal part of the Striatal Bundle (StB_t_). Interestingly, for the HCP data, the temporal part of the Extreme Capsule (EmC_t_) and the Muratoff Bundle were also the tracts with the lowest/highest correlations, respectively. Hence, certain tract reconstructions were consistently more variable than others across subjects, which may hint to also being more challenging to reconstruct. We have now clarified these aspects in the corresponding Results section.

(11) Anatomical validation of the new cortical-subcortical tractsI really appreciated the use of prior tract tracing findings to anatomically validate the corticalsubcortical tractography outputs for both the cortical-striatal and amygdalofugal tracts. It struck me, however, that the anatomical validation was purely qualitative, focused on the relative positioning or the topographical organization of major connections. The anatomical validation would be strengthened if profiles of connectivity between cortical regions and specific subcortical nuclei or subcortical subdivisions could be quantitatively compared, if at all possible. Can the differential connectivity shown visually for the putamen in Figure 3 be quantified for the tract tracing data and the tractography outputs? Does the amygdalofugal bundle show differential/preferential connectivity across amygdala nuclei in tract tracing data, and is this seen in tractography?

We appreciate the comment, please see Reply to your comment 6 above. In addiIon to the challenges described there, we do not have access to terminal fields other than in the striatum and these ones are 2D, so we make a qualitaIve comparison of the relevant connecIvity contrasts. We expect that a number of currently ongoing high-profile BRAIN CONNECTS Centres (such as the LINC and the CMC) will be addressing such challenges in the coming years and will provide the tools and data to the community to perform such quanItaIve comparisons at scale.

(12) I believe that all visualizations of the macaque and human tractography showed groupaveraged maps. What do these tracts look like at the individual level? Understanding individual-level performance and anatomical variation is important, given the Discussion paragraph on using this method to guide neuromodulation.

We now demonstrate some representative examples of individual subject reconstructions in Supplementary Figures S6, S7, ranking subjects by the average agreement of individual tract reconstructions to the mean and depicting the 10th percentile, median and 90th percentile of these subjects. We have also shown more results in Author response images 1-2, generated by TractSeg, to indicate how a different bundle segmentation approach would handle individual variability compared to our approach.

(13) Connectivity-based comparisons across species:Figures 5 and 6 of the manuscript show that, as compared to using only cortico-cortical XTRACT tracts, using the full set of XTRACT tracts (with new cortical-subcortical tracts) allows for more specific mapping of homologous subcortical and cortical regions across humans and macaques. Is it possible that this result is driven by the fact that the "connectivity blueprints" for the subcortex did not use an intermediary GM x WM matrix to identify connection patterns, whereas the connectivity blueprints for the cortex did? I was surprised that a whole brain GM x WM connectivity matrix was used in the cortical connectivity mapping procedure, given known problems with false positives etc., when doing whole brain tractography - especially aHer such anatomical detail was considered when deriving the original tracts. Perhaps the intermediary step lowers connectivity specificity and accuracy overall (as per Figure 9), accounting for the poorer performance for cortico-cortical tracts?

The point is well-taken, however it cannot drive the results in Figures 5 and 6. Before explaining this further, let us clarify the raIonale of using the GMxWM connecIvity matrix, which we have published quite extensively in the past for cortico-cortical connecIons (Mars, eLife 2018 - Warrington, Neuroimage 2020 - Roumazeilles, PLoS Biology 2020 - Warrington, Science Advances 2022 – Bryant, J Neuroscience 2025).

Having established the bodies of the tract using the XTRACT protocols, we use this intermediate step of multiplying with a GM x WM connectivity matrix to estimate the grey matter projections of the tracts. The most obvious approach of tracking towards the grey matter (i.e. simply find where tracts intersect GM) has the problem that one moves through bottlenecks in the cortical gyrus and after which fibres fan out. Most tractography algorithms have problems resolving this fanning. However, we take the opposite approach of tracking from the grey matter surface towards the white matter (GMxWM connectivity matrix), thus following the direction in which the fibres are expected to merge, rather than to fan out. We then multiply the GMxWM tractrogram with that of the body of the tract to identify the grey matter endpoints of the tract. This avoids some of the major problems associated with tracking towards the surface. In fact, using this approach improves connectivity specificity towards the cortex, rather than the opposite. We provide some indicative results here for a few tracts:

**Author response image 5. sa3fig5:** Connectivity profiles for example cortico-cortical tracts with and without using the intermediary GMxWM matrix. Tracts considered are the Superior Longitudinal Fasciculus 1 (SLF_1_), Superior Longitudinal Fasciculus 2 (SLF_2_), the Frontal Aslant (FA) and the Inferior Fronto-Occipital Fasciculus (IFO). We see that the surface connectivity patterns without using the GMxWM intermediary matrix are more diffuse (effect of “fanning out” gyral bias), with reduced specificity, compared to whenusing the GMxWM matrix

Tracking to/from subcortical nuclei does not have the same tractography challenges as tracking towards the cortex and in fact we found that using the intermediary GMxWM matrix is less favourable for subcortex (Figure 9), which is why we opted for not using it.

Regardless of how cortical and subcortical connectivity patterns are obtained, the results in Figures 5 and 6 utilise only cortical connectivity patterns. Hence, no matter what tracts are considered (cortico-cortical or cortico-subcortical) to build the connectivity patterns, these results have been obtained by always using the intermediate step of multiplying with the GMxWM connectivity matrix (i.e. it is not the case that cortical features are obtained with the intermediate step and subcortical features without, all of them have the intermediate step applied, as the connectivity patterns comprise of cortical endpoints). Figure 9 is only applicable for subcortical endpoints that play no role in the comparisons shown in Figures 5 and 6. We hope this clarifies this point.

(14) Methodological clarifications:The Methods describe how anatomical masks used in tractography were delineated in standard macaque space and then translated to humans using "correspondingly defined landmarks". Can the authors elaborate as to how this translation from macaques to humans was accomplished?

For a given tract, our process for building a protocol involved looking into the wider anatomical literature, including the standard white matter atlas of Schmahmann and Pandya (2006) and numerous anatomy papers that are referenced in the protocol description, to determine the expected path the tract was meant to take in white matter and which cortical and subcortical regions are connected. This helped us define constraints and subsequently the corresponding masks. The masks were created through the combination of hand-drawn ROIs and standard space atlases. We firstly started with the macaque where tracer literature is more abundant, but, importantly, our protocol definitions have been designed such that the same protocol can be applied to the human and macaque brain. All choices were made with this aspect in mind, hence corresponding landmarks between the two brains were considered in the mask definition (for instance “the putamen”, “a sub-commissural white matter mask”, the “whole frontal pole” etc, as described in the protocol descriptions).

The protocols have not been created by a single expert but have been collated from multiple experts (co-authors SA, SW, DF, KB, SH, SS drove this aspect) and the final definitions have been agreed upon by the authors.

(15) The article heavily utilizes spatial path distribution maps/normalized path distributions, yet does not describe precisely what these are and how they were generated. Can the authors provide more detail, along with the rationale for using these with Pearson's correlations to compare tracts across subjects (as opposed to, e.g., overlap sensitivity/specificity or the Jaccard coefficient)?

We have now clarified in text how these plots are generated, particularly when compared using correlation values. We tried Jaccard indices on binarized masks of the tracts and these gave similar trends to the correlations reported in Figure 4 (i.e. higher similarities within that across cohorts). We however feel that correlations are better than Jaccard indices, as the latter assume binary masks, so they focus on spatial overlap ignoring the actual values of the path distributions, we hence kept correlations in the paper.

**Reviewing Editor Comments**
“The reviewers had broadly convergent comments and were enthusiastic about the work. As further detailed by Reviewer 3 (see below), if the authors choose to pursue revisions, there are several elements that have the potential to enhance impact.”

Thank you, we have replied accordingly and aimed to address most of the comments of the Reviewers.

“Comparison to existing methods. How does this approach compare to other approaches cited by the authors?”

Please see replies to Comment 2 of Reviewer 1 and Comment 7 of Reviewer 2. Briefly, we have now generated new results and clarified aspects in the text.

“Minimum data requirements. How broadly can this approach be used across scan variation? How does this impact data from individual participants? Displaying individual participants may help, in addition to group maps.”

Please see replies to Comment 10 of Reviewer2 on minimum data requirements and individual parIcipants, as well as to Comment 3 of Reviewer 1 on the actual groups considered. Briefly, we have generated new figures and regenerated results using UKBiobank data.

Softare. What are the sofware requirements? Is the approach interoperable with other methods?”

Please see Reply to Comment 9 of Reviewer 2. Our protocols can be used to guide tractography using other types of data as they comprise of guiding ROIs for a given tract. So, although we have not tested them beyond FSL-XTRACT, we believe they can be useful with other tractography packages as well, as there is nothing FSL-specific in these anatomically-informed recipes.

“Comparisons with tract tracing. To the degree possible, quantitative comparisons with tract tracing data would bolster confidence in the method.”

Please see Replies to Comments 6 and 11 of Reviewer 2. Briefly, we appreciate the comment and it is something we would love to do, but there are no data readily available that would allow such quanItaIve comparison in a meaningful way. This is a known challenge in the tractography field, which is why NIH has invested in two 5 year Centres to address it. Our approach will provide a solid starIng point for opImising and comparing further cortico-subcortical tractography reconstructions against microscopy and tracers in the same animal and at scale.